# MAVIS:
# MATHEMATICAL VISUAL INSTRUCTION TUNING WITH AN AUTOMATIC DATA ENGINE

**Renrui Zhang**[1*†]**, Xinyu Wei**[3*]**, Dongzhi Jiang**[1]**, Ziyu Guo**[2]**, Yichi Zhang**[3]**, Chengzhuo Tong**[4]
**Jiaming Liu**[3]**, Aojun Zhou**[1]**, Shanghang Zhang**[3]**, Peng Gao**[4]**, Hongsheng Li**[1,5‡]

[1]CUHK MMLab  &  [2]MiuLar Lab    [3]Peking University
[4]Shanghai AI Laboratory    [5]CPII under InnoHK
renruizhang@link.cuhk.edu.hk, allen_wei@stu.pku.edu.cn

[*] Equal contribution   [†] Project lead   [‡] Corresponding author

## ABSTRACT

Multi-modal Large Language Models (MLLMs) have recently showcased superior proficiency in general visual scenarios. However, we identify their mathematical capabilities remain under-explored with three areas to be improved: *visual encoding of math diagrams*, *diagram-language alignment*, and *chain-of-thought (CoT) reasoning*. This draws forth an urgent demand for an effective training paradigm and a large-scale, comprehensive dataset with detailed CoT rationales, which is challenging to collect and costly to annotate manually. To tackle this issue, we propose **MAVIS**, a **MA**thematical **VIS**ual instruction tuning pipeline for MLLMs, featuring an automatic data engine to efficiently create mathematical visual datasets. We design the data generation process to be entirely independent of human intervention or GPT API usage, while ensuring the diagram-caption correspondence, question-answer correctness, and CoT reasoning quality. With this approach, we curate two datasets, MAVIS-Caption (558K diagram-caption pairs) and MAVIS-Instruct (834K visual math problems with CoT rationales), and propose four progressive stages for training MLLMs from scratch. First, we utilize MAVIS-Caption to fine-tune a math-specific vision encoder (CLIP-Math) through contrastive learning, tailored for improved diagram visual encoding. Second, we also leverage MAVIS-Caption to align the CLIP-Math with a large language model (LLM) by a projection layer, enhancing vision-language alignment in mathematical domains. Third, we adopt MAVIS-Instruct to perform the instruction tuning for robust problem-solving skills, and term the resulting model as MAVIS-7B. Fourth, we apply Direct Preference Optimization (DPO) to enhance the CoT capabilities of our model, further refining its step-wise reasoning performance. On various mathematical benchmarks, our MAVIS-7B achieves leading results among open-source MLLMs, e.g., surpassing other 7B models by +9.3% and the second-best LLaVA-NeXT (110B) by +6.9%, demonstrating the effectiveness of our method. Data and models are released at https://github.com/ZrrSkywalker/MAVIS.

## 1 INTRODUCTION

The pursuit of artificial general intelligence necessitates models to seamlessly interpret and generate multi-modal data. In recent years, the advent of Large-language Models (LLMs) (Brown et al., 2020; Touvron et al., 2023a;b; Chiang et al., 2023) and their Multi-modal extension (MLLMs) (Zhang et al., 2024a; Gao et al., 2023b; Su et al., 2023; Ye et al., 2023a) have significantly facilitated this process across various fields, such as healthcare (Singhal et al., 2023; Shu et al., 2023), autonomous driving (Yang et al., 2023; Jin et al., 2024), and robotics (Li et al., 2023b; Liu et al., 2024b). Although MLLMs exhibit remarkable performance in diverse tasks and benchmarks, one arena where they have yet to fully demonstrate their potential is mathematical problem-solving in visual contexts.

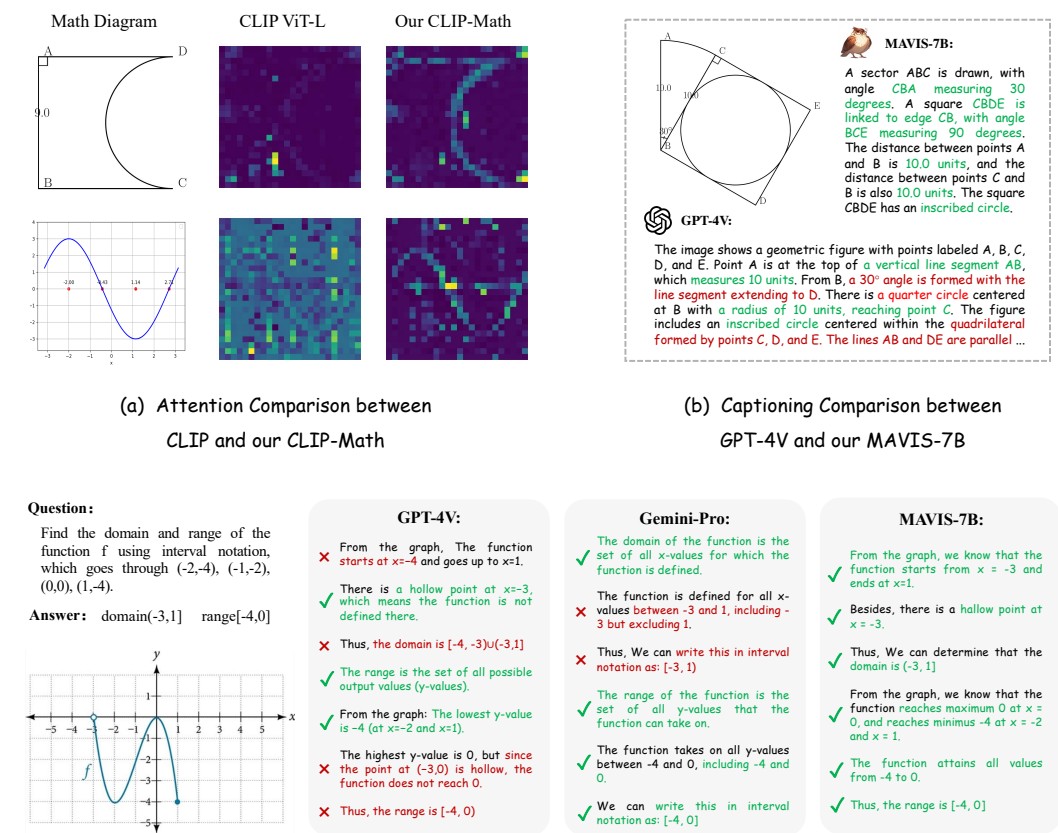

Figure 1: **(a)** We compare the attention map of class tokens from CLIP ViT-L (Radford et al., 2021) and our CLIP-Math. Our vision encoder can better capture significant mathematical information within diagrams. **(b)** We compare the diagram captioning capabilities between GPT-4V (OpenAI, 2023c) and our MAVIS-7B, where GPT-4V fall short of accurately recognizing mathematical elements. **(c)** We compare the chain-of-thought (CoT) reasoning between different models, showcasing that GPT-4V and Gemini-Pro (Gemini Team, 2023) suffer from low-quality reasoning process.

Existing efforts (OpenAI, 2023b;a; Zhou et al., 2023) for text-only mathematics have attained considerable progress, largely attributed to the availability of sufficient and easily accessible training data. In contrast, solving visual mathematical problems remains a significant challenge for MLLMs, primarily due to the absence of a fully validated, effective training pipeline and the acute shortage of large-scale, high-quality datasets. Visual mathematical data is not only more costly to collect from publicly available sources compared to text-only data, but also requires expensive manual annotation to produce accurate step-by-step chain-of-thought (CoT) rationales integrating diagram information. In light of these challenges, we identify three critical issues that impede the visual mathematical capabilities of MLLMs.

  i. **Unsatisfactory math diagram embeddings by vision encoders.** Most MLLMs adopt a frozen CLIP (Radford et al., 2021) as the vision encoder, which is pre-trained by natural images capturing real-world scenes with rich colors and textures. In contrast, math diagrams are composed of abstract curves, shapes, and symbols with a monochromatic color scheme, exhibiting large semantic gaps to general scenarios. As visualized in Figure 1 (a), the attention map of CLIP struggles to capture important information within math diagrams, which cannot provide satisfactory visual embeddings for LLMs to understand.

  ii. **Diagram-language misalignment between vision encoders and LLMs.** Likewise, the vision-language pre-training stage of MLLMs also adopts natural image-caption pairs for cross-modal alignment. Due to the domain gap, while they can generate accurate captions for

real-world images, but fall short of recognizing basic mathematical elements and narrating their relations. As compared in Figure 1 (b), even GPT-4V (OpenAI, 2023c) produces low-quality descriptions for simple geometric figures and functions, indicating LLMs are not well aligned with the visual embedding space of math diagrams.

iii. **Inaccurate CoT reasoning capabilities with visual elements by MLLMs.** Referring to the CoT evaluation in MathVerse (Zhang et al., 2024b), incorporating the diagram input would adversely affect the reasoning quality of MLLMs compared to using only the text-only question. As visualized in Figure 1 (c), we observe the problem-solving process of GPT-4V and Gemini-Pro (Gemini Team, 2023) both suffer from low-quality CoT reasoning accuracy. This demonstrates the incapability of MLLMs to leverage visual cues for precise step-by-step mathematical problem-solving.

Therefore, to mitigate these issues, it is essential to develop an extensive dataset and effective training approach tailored to visual mathematics. In this paper, we propose **MAVIS**, a **MA**thematical **VIS**ual instruction tuning paradigm and an automatic data generation engine for MLLMs, which aims to fully unleash their potential for diagram visual encoding and reasoning capabilities. We introduce two meticulously curated datasets, a progressive four-stage training pipeline, and a visual mathematical specialist, MAVIS-7B. We summarize the contributions of our work as follows.

- **Automatic Mathematical Visual Data Engine.** To eliminate the need for labor-intensive annotation and expensive GPT API (OpenAI, 2023c;b) usage, we designed our data engine to be entirely rule-based and fully automated. This engine handles every aspect of mathematical data creation, including diagram drawing, caption generation, question-answer synthesis, and CoT rationale production. With this approach, we curate two large-scale, high-quality mathematical visual datasets, MAVIS-Caption and MAVIS-Instruct, widely covering plane geometry, analytic geometry, and function. MAVIS-Caption consists of 558K diagram-caption pairs automatically created by our data engine with accurate vision-language correspondence. MAVIS-Instruct includes 834K visual math problems, which includes 582K data constructed by our data engine and additional 252K data augmented by GPT-4V from manual collection and existing datasets (Chen et al., 2021c; Lu et al., 2021). Each problem is annotated with a CoT rationale, and modified to contain minimized textual redundancy that enforces MLLMs to pay more attention on visual diagrams.

- **Four-stage Training Pipeline.** Our training framework involves four progressive stages designed to sequentially address the aforementioned identified deficiencies in MLLMs. Firstly, we utilize MAVIS-Caption to fine-tune a math-specific vision encoder by contrastive learning, termed CLIP-Math, to enable better visual representations of math diagrams. Subsequently, we align this encoder with the LLM to ensure effective diagram-language integration also by MAVIS-Caption. After that, our MAVIS-Instruct is adopted to instruction-tune the MLLM, which provides sufficient step-wise problem-solving supervision. Finally, we employ Direct Preference Optimization (DPO) (Rafailov et al., 2024) with annotated CoT rationales in MAVIS-Instruct to further enhance the reasoning capabilities of our model.

- **Mathematical Visual Specialist.** After the four-stage training, we develop MAVIS-7B, an MLLM specifically optimized for visual mathematical problem-solving. On various evaluation benchmarks, our model achieves leading performance compared to existing open-source MLLMs, e.g., surpassing other 7B models by +9.3% and the second-best LLaVA-NeXT (110B) (Li et al., 2024a) by +6.9% on MathVerse (Zhang et al., 2024b). The quantitative results and qualitative analysis both validate the significance of our approach.

## 2 AUTOMATIC DATA ENGINE

To cope with the substantial data requirements of MLLMs, it is essential to have access to extensive training instances. However, for visual mathematics, the paucity of publicly available datasets poses a challenge, and creating such data manually also involves a high cost. Therefore, as illustrated in Figure 2, we develop an automatic data engine to efficiently generate high-quality math diagrams (Section 2.1), captions (Section 2.2), and question-answer with rationales (Section 2.3).

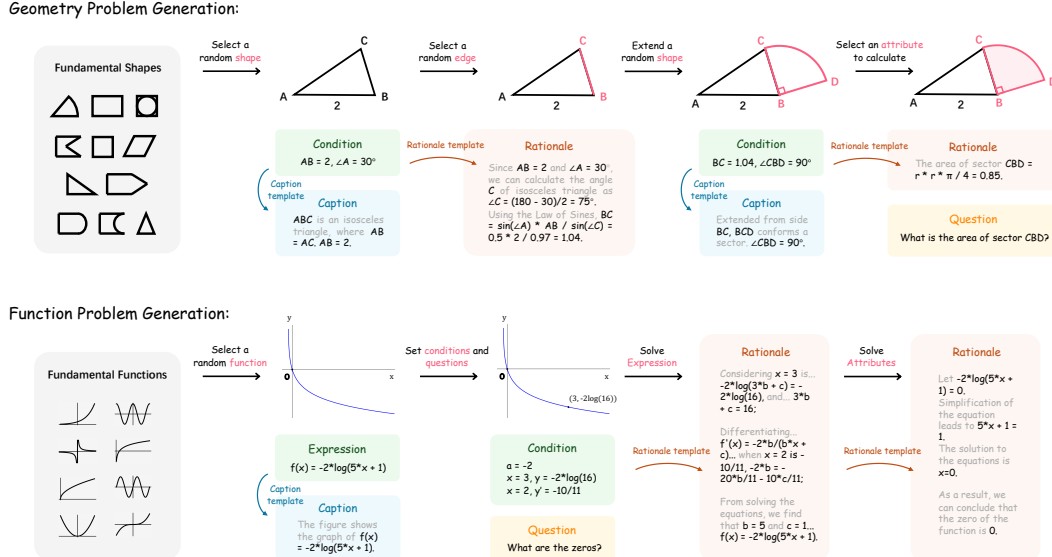

Figure 2: **Overview of Automatic Data Engine.** We present the generation pipelines of geometry (Top) and function (Bottom) problems within the proposed automatic data engine, including diagrams, questions, captions, and Chain-of-Thought (CoT) rationales.

## 2.1 DIAGRAM GENERATION

Covering most mathematical scenarios, we adopt three diagram types: plane geometry, analytic geometry, and function. Note that all the logic of the data engine is implemented in Python, and we employ Matplotlib for the graphical rendering of the diagrams.

**Plane Geometry Diagram.** As such diagrams typically consist of spatial combinations of various basic shapes, we utilize principles from multi-hop data curation to develop customized generation rules. These rules allow for the iterative integration of new shapes into existing configurations. Initially, we establish a core set of shapes, including squares, rectangles, triangles, sectors, etc, for diagram generation. Starting with a randomly selected shape, we extend another shape from the set along one of its straight sides. By iterating this process, we can construct diverse plane geometry diagrams featuring different combinations of shapes. Additionally, we randomly label the vertices with letters (e.g., A, B, C) and annotate numerical values relevant to geometric properties (e.g., side lengths and angles), simulating realistic plane geometry problems.

**Analytic Geometry Diagram.** Likewise, our approach begins by defining a basic figure set that differs slightly from that used in plane geometry; for example, we include additional elements such as points and line segments. We then construct a Cartesian coordinate system, complete with grid lines and scaled axes. The range of the coordinate system is randomly determined within a predefined scope. Subsequently, we select a number from 1 to 3 to indicate the number of figures to be drawn on the graph, and randomly choose coordinates for the top-left vertices to plot these figures at varied sizes (using these points as centers for circles). Unlike plane geometry, we ensure that the figures do not overlap, except for points and segments, and maintain the figure areas within a suitable scale.

**Function Diagram.** We focus on seven fundamental function types: polynomial, sine, cosine, tangent, logarithmic, absolute value, and piece-wise polynomial functions. For each function type, we parameterize the equations with random variables, such as coefficients and constants within a predefined range (e.g., $a$ and $b$ in $y = ax + b$), which facilitates the generation of diverse function graphs. We also adopt the same Cartesian coordinate system employed for analytic geometry. Additionally, for specific caption or question-answering samples, we also plot key features like extreme points and zero points of the functions, providing additional visual information that aids in the understanding and reasoning of these mathematical functions.

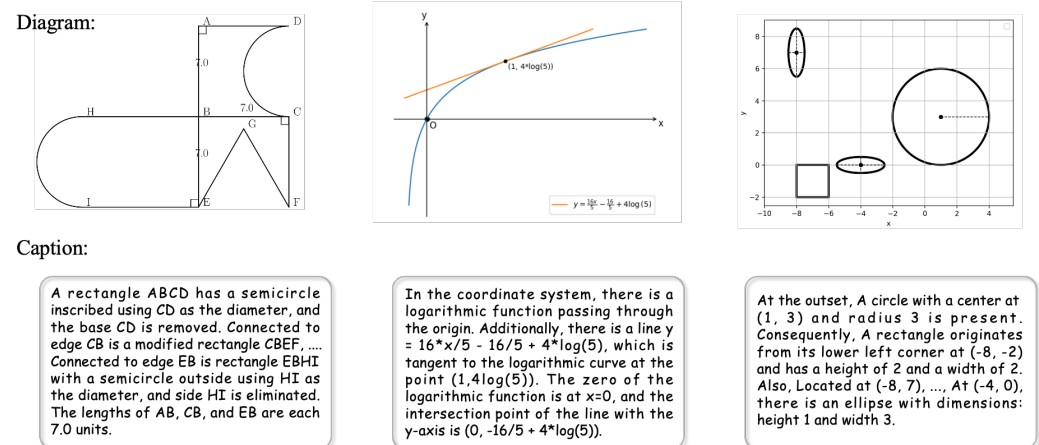

**Figure 3: MAVIS-Caption Dataset.** We showcase three diagram-caption pairs of plane geometry, function, and analytic geometry in MAVIS-Caption, generated by our developed data engine.

## 2.2 MAVIS-CAPTION

With our mathematical visual data engine, we first curate a diagram-caption dataset, MAVIS-Caption, as shown in Figure 3, aiming to benefit the diagram visual representations and cross-modal alignment.

**Data Overview.** As presented in Table 3 of the Appendix, the MAVIS-Caption dataset comprises 588K diagram-caption pairs. This includes 299K for plane geometry, 77K for analytic geometry, and 212K for function. The average word length of the captions is 61.48 words, reflecting their detailed descriptive nature. The overall vocabulary size is 149, indicating the diversity in language expression. We adopt different strategies to generate captions for three types of diagrams. It is important to note that GPT-4 (OpenAI, 2023b) is only utilized during the template creation stage; it is not used at any point during the automatic caption generation process.

**Plane Geometry Caption.** We follow the iterative geometric generation process to develop regulations for an accurate and detailed caption. We first prompt GPT-4 to create three sets of language templates: the descriptive content for fundamental shapes (e.g., *"A Triangle {} with two congruent sides {} and {}"*), the phrases to denote specific attributes (e.g., *"Angle {} measures {} degrees"*), and the conjunction to link two adjacent shapes (e.g., *"Attached to edge {} of shape {}, there is a {}"*). Then, based on various generation scenarios, we fill and merge these templates to acquire a coherent description of the geometric figure.

**Function Caption.** As function diagrams typically showcase a single curve, we directly utilize GPT-4 to generate templates describing various properties of functions, including expressions, domains, ranges, extreme points, and zero points. Each template is then filled based on specific cases, such as *"The expression of the function is $y = -3x^3 - 2x^2 - 2x - 2$. Within the range of x values $[-3.0, 4.0]$, zero points occur at $-0.83$ ..."*.

**Analytic Geometry Caption.** We also employ GPT-4 to obtain two sets of language templates: the description of coordinates and attribute information for basic figures (e.g., *"The square with its base left corner at {} features sides of {} in length"*) and the spatial relation for nearby figures (e.g., *"On the bottom right of {}, there is a {}"*). The captions are then formulated by filling in the coordinates and selecting appropriate spatial relationship templates through coordinate comparison.

## 2.3 MAVIS-INSTRUCT

Besides the diagram-caption data, we curate MAVIS-Instruct of extensive problem-solving data, which endows MLLMs with visual mathematical reasoning capabilities and serve as the basis for Direct Preference Optimization (DPO) (Rafailov et al., 2024), as shown in Figure 5.

Diagram:

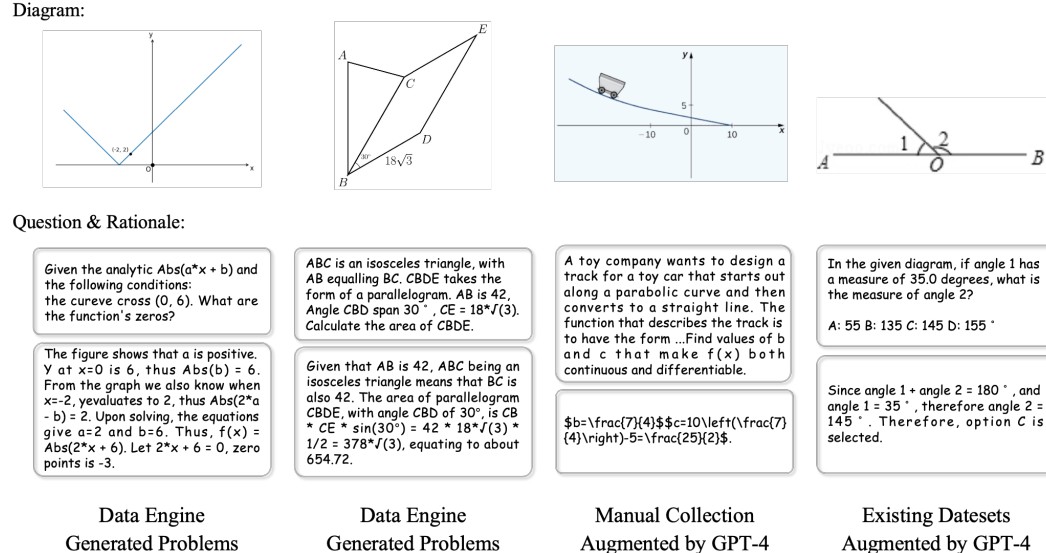

Question & Rationale:

| | | | |
|---|---|---|---|
| Given the analytic Abs(a*x + b) and the following conditions: the curve cross (0, 6). What are the function's zeros? | ABC is an isosceles triangle, with AB equalling BC. CBDE takes the form of a parallelogram. AB is 42, Angle CBD span 30˚, CE = 18*√(3). Calculate the area of CBDE. | A toy company wants to design a track for a toy car that starts out along a parabolic curve and then converts to a straight line. The function that describes the track is to have the form ...Find values of b and c that make f(x) both continuous and differentiable. | In the given diagram, if angle 1 has a measure of 35.0 degrees, what is the measure of angle 2? A: 55 B: 135 C: 145 D: 155˚ |
| The figure shows that a is positive. Y at x=0 is 6, thus Abs(b) = 6. From the graph we also know when x=-2, yevaluates to 2, thus Abs(2*a - b) = 2. Upon solving, the equations give a=2 and b=6. Thus, f(x) = Abs(2*x + 6). Let 2*x + 6 = 0, zero points is -3. | Given that AB is 42, ABC being an isosceles triangle means that BC is also 42. The area of parallelogram CBDE, with angle CBD of 30°, is CB * CE * sin(30°) = 42 * 18*√(3) * 1/2 = 378*√(3), equating to about 654.72. | $b=\frac{7}{4}$$c=10\left(\frac{7}{4}\right)-5=\frac{25}{2}$. | Since angle 1 + angle 2 = 180˚, and angle 1 = 35˚, therefore angle 2 = 145˚. Therefore, option C is selected. |
| Data Engine Generated Problems | Data Engine Generated Problems | Manual Collection Augmented by GPT-4 | Existing Datesets Augmented by GPT-4 |

Figure 4: **MAVIS-Instruct Dataset.** We showcase the generated visual math problems from four sources within MAVIS-Instruct, which contain detailed rationales and minimized textual redundancy.

**Data Overview.** As illustrated in Table 4 of the Appendix, the MAVIS-Instruct dataset consists of a total of 834K visual math problems. Given that the proportion of analytic geometry problems is relatively small, we classify them with function problems for simplicity. Each problem in MAVIS-Instruct includes a CoT rationale providing step-by-step solutions, with an average answer length of 150 words. We have minimized textual redundancy in the questions, eliminating unnecessary contextual information, distracting conditions, and attributes readily observable from the diagrams. This reduction in text forces MLLMs to enhance their capability to extract essential content from visual inputs. MAVIS-Instruct is assembled from four distinct sources to ensure broad coverage.

**Data Engine Generated Problems.** Within our data engine, we manually craft rigorous regulations to produce visual math problems with accurate CoT annotations. Similar to caption generation, GPT API is not involved in the automatic synthesis process of questions, answers, and CoT rationales.

- **Plane Geometry Problems.** We initially prompt GPT-4 to compile a comprehensive set of mathematical formulas applicable to each basic shape (e.g., Pythagorean theorem for right triangles and area formula for circles). Then, for a geometric diagram, we randomly select a known condition within a shape as the final solution target, and systematically deduce backward to another condition, either within the same shape or an adjacent one, using a randomly selected mathematical formula. This deduced condition is then set as unknown, and we continue iterative backward deductions as necessary. The final condition, along with any conditions in the last step, are presented as initial attributes in the question. The rationales can be simply obtained by reversing this backward deduction process.

- **Function Problems.** As the properties of functions are predetermined, we utilize GPT-4 to generate diverse reasoning templates. These templates facilitate the solving of one function property based on other provided properties, thereby ensuring the generation of high-quality function rationales. The related function properties include analytical expression, function values, zeros, extremum points, monotonicity, derivatives, and integrals. To accurately reason these properties, the CoT annotation incorporates understanding of function types, solving the analytical expressions of equations, and interpreting function graphs.

**Data Engine Captions Annotated by GPT-4.** Given the detailed captions and diagrams generated by our data engine, we can prompt GPT-4V with these sufficient conditions to synthesis question-answering data and ensure its correctness. We first generate a new set of 17K diagram-caption pairs that do not overlap with the previous MAVIS-Caption, which avoids answer leakage within the detailed caption. Then, we prompt GPT-4V to generate 3 new problems with rationales, obtaining 51K data in total from the diagram-caption pairs.

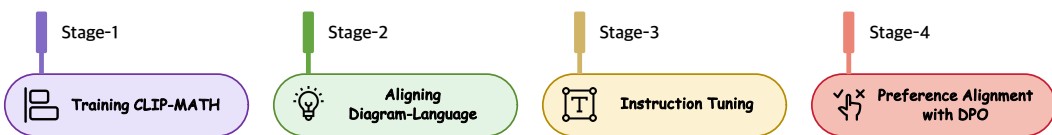

Figure 5: **Four-stage Training Pipeline of MAVIS.** With our curated MAVIS-Caption and MAVIS-Instruct, we adopt four progressive stages for training a mathematical visual specialist from scratch.

**Manual Collection Augmented by GPT-4.**   To incorporate high-quality problems found in real-world contexts, we manually collect 4K math problems with diagrams from publicly available resources. Recognizing that these sources often lack detailed rationales and may contain redundant text, we initially utilize GPT-4V to annotate a detailed solving process and streamline the question text to reduce redundancy. Subsequently, for each collected instance, we input the question, rationale, and diagram into GPT-4 and employ customized few-shot prompts to generate 20 new problems per original, comprising 15 multiple-choice questions and 5 free-form questions. This process contributes a total of 83K problems to the dataset.

**Existing Datasets Augmented by GPT-4.**   Given existing well-organized geometric datasets, we can also leverage them to expand MAVIS-Instruct. Referring to previous prompt designs, we augment the 8K training set from two dataset, Geometry-3K (Lu et al., 2021) and GeoQA+ (Chen et al., 2021b), into 80K visual problems with accompanying rationales, mapping each original problem to 10 new ones. Due to the scarcity of publicly available function data, we do not include function problems from this source.

## 3   Mathematical Visual Training

With the curated datasets, we devise a four-stage training pipeline for endowing MLLMs with mathematical visual capabilities. They respectively aim to mitigate the three deficiencies within existing MLLMs, i.e., diagram visual encoding, diagram-language alignment, and mathematical reasoning skills in visual contexts.

### 3.1   Stage 1: Training CLIP-Math

To enhance CLIP's (Radford et al., 2021) inadequate visual encoding of math diagrams, we utilize MAVIS-Caption to train a specialized CLIP-Math encoder. Specifically, we fine-tune a pre-trained CLIP-Base model following the conservative learning scheme. The math diagrams are fed into the learnable vision encoder, while the corresponding captions are processed by the text encoder, which remains frozen to provide reliable supervision. Via contrastive training, the model learns to adapt from its original natural image domain to mathematical contexts, increasing its focus on essential visual elements within diagrams, as demonstrated in Figure 1 (a). The optimized CLIP-Math encoder now delivers more precise and robust representations of math diagrams, establishing a solid foundation for the subsequent visual interpretation of LLMs.

### 3.2   Stage 2: Aligning Diagram-language

After acquiring the CLIP-Math encoder, we further integrate it with LLMs using MAVIS-Caption to boost cross-modal alignment between math diagrams and language embedding space. Using a simple two-layer MLP as the projection layer, we transform the visual encodings from CLIP-Math, and prepend them as a prefix to the LLM input. This process, guided by the diagram captioning task, enables the LLM to accurately recognize mathematical components and spatial arrangements. With the diagram-language alignment, LLMs are equipped with the interpretation capability in math diagrams, serving as an initial step toward deeper mathematical reasoning. In this stage, we freeze the CLIP-Math, and train the projection layer along with the LoRA-based (Hu et al., 2021) LLM.

### 3.3   Stage 3: Instruction Tuning

On top of that, we leverage MAVIS-Instruct to endow MLLMs with CoT reasoning and problem-solving capabilities in visual mathematics. The detailed rationales within each problem's solution

Table 1: **Evaluation on MathVerse's *testmini* Set with Six Problem Versions.** 'CoT-E' and 'Acc' denote the scores of CoT evaluation strategy and the scores of direct 'true or false' accuracy, respectively. '*' denotes previous mathematical visual specialists. The highest scores for closed-source and open-source MLLMs are marked in red and blue, respectively.

| Model | LLM Size | All | | Text Dominant | | Text Lite | | Vision Intensive | | Vision Dominant | | Vision Only | |
|---|---|---|---|---|---|---|---|---|---|---|---|---|---|
| | | CoT-E | Acc | CoT-E | Acc | CoT-E | Acc | CoT-E | Acc | CoT-E | Acc | CoT-E | Acc |
| *Baselines* | | | | | | | | | | | | | |
| Random Chance | - | - | 12.4 | - | 12.4 | - | 12.4 | - | 12.4 | - | 12.4 | - | 12.4 |
| Human | - | - | 64.9 | - | 71.2 | - | 70.9 | - | 61.4 | - | 68.3 | - | 66.7 |
| *LLMs* | | | | | | | | | | | | | |
| ChatGPT | - | - | - | 51.3 | 33.3 | 38.5 | 18.9 | - | - | - | - | - | - |
| GPT-4 | - | - | - | 63.4 | 46.5 | 40.7 | 20.7 | - | - | - | - | - | - |
| *Closed-source MLLMs* | | | | | | | | | | | | | |
| Qwen-VL-Plus | - | 21.3 | 11.8 | 26.0 | 15.7 | 21.2 | 11.1 | 18.5 | 9.0 | 19.1 | 13.0 | 21.8 | 10.0 |
| Gemini-Pro | - | 35.3 | 23.5 | 39.8 | 26.3 | 34.7 | 23.5 | 32.0 | 23.0 | 36.8 | 22.3 | 33.3 | 22.2 |
| Qwen-VL-Max | - | 37.2 | 25.3 | 42.8 | 30.7 | 37.7 | 26.1 | 33.6 | 24.1 | 35.9 | 24.1 | 35.9 | 21.4 |
| GPT-4V | - | 54.4 | 39.4 | 63.1 | 54.7 | 56.6 | 41.4 | 51.4 | 34.9 | 50.8 | 34.4 | 50.3 | 31.6 |
| *Open-source MLLMs* | | | | | | | | | | | | | |
| LLaMA-Adapter-V2 | 7B | 5.8 | 5.7 | 7.8 | 6.2 | 6.3 | 5.9 | 6.2 | 6.1 | 4.5 | 4.2 | 4.4 | 6.1 |
| ImageBind-LLM | 7B | 10.0 | 9.2 | 13.2 | 11.4 | 11.6 | 11.3 | 9.8 | 8.9 | 11.8 | 11.2 | 3.5 | 3.4 |
| mPLUG-Owl2 | 7B | 10.3 | 5.9 | 11.6 | 6.6 | 11.4 | 6.3 | 11.1 | 6.3 | 9.4 | 5.6 | 8.0 | 4.9 |
| MiniGPT-v2 | 7B | 10.9 | 11.0 | 13.2 | 12.1 | 12.7 | 12.0 | 11.1 | 13.1 | 11.3 | 10.3 | 6.4 | 7.4 |
| LLaVA-1.5 | 7B | 12.7 | 7.6 | 17.1 | 8.8 | 12.0 | 7.6 | 12.6 | 7.4 | 12.7 | 7.4 | 9.0 | 6.9 |
| SPHINX-Plus | 13B | 14.0 | 12.2 | 16.3 | 13.9 | 12.8 | 11.6 | 12.9 | 11.6 | 14.7 | 13.5 | 13.2 | 10.4 |
| G-LLaVA* | 7B | 15.7 | 16.6 | 22.2 | 20.9 | 20.4 | 20.7 | 16.5 | 17.2 | 12.7 | 14.6 | 6.6 | 9.4 |
| LLaVA-NeXT | 8B | 17.2 | 15.6 | 21.6 | 19.4 | 19.7 | 15.2 | 17.6 | 16.8 | 14.9 | 15.2 | 12.1 | 11.3 |
| ShareGPT4V | 13B | 17.4 | 13.1 | 21.8 | 16.2 | 20.6 | 16.2 | 18.6 | 15.5 | 16.2 | 13.8 | 9.7 | 3.7 |
| SPHINX-MoE | 8×7B | 22.8 | 15.0 | 33.3 | 22.2 | 21.9 | 16.4 | 21.1 | 14.8 | 19.6 | 12.6 | 18.3 | 9.1 |
| Math-LLaVA* | 13B | 24.1 | 19.0 | 34.2 | 21.2 | 22.7 | 19.8 | 21.1 | 20.2 | 20.3 | 17.6 | 22.2 | 16.4 |
| InternLM-XC2. | 7B | 25.9 | 16.5 | 36.9 | 22.3 | 28.3 | 17.0 | 20.1 | 15.7 | 24.4 | 16.4 | 19.8 | 11.0 |
| LLaVA-NeXT | 110B | 28.3 | 24.5 | 37.1 | 31.7 | 29.1 | 24.1 | 22.6 | 21.0 | 21.8 | 22.1 | 30.9 | 20.7 |
| **MAVIS-7B w/o DPO*** | 7B | 33.7 | 27.5 | 42.5 | 41.4 | 36.3 | 29.1 | 33.3 | 27.4 | 29.3 | 24.9 | 27.1 | 14.6 |
| **MAVIS-7B*** | 7B | 35.2 | 28.4 | 43.2 | 41.6 | 37.2 | 29.5 | 34.1 | 27.9 | 29.7 | 24.7 | 31.8 | 18.3 |

provide high-quality reasoning guidance for MLLMs, significantly enhancing their step-by-step CoT process. Furthermore, as we have minimized the redundancy within question texts during the construction process, such text-lite problem formats, referring to MathVerse (Zhang et al., 2024b), facilitate MLLMs to capture more essential information from the visual embeddings for problem-solving, rather than relying on shortcuts to only process the textual content. During this stage, we unfreeze both the projection layer and the LoRA-based (Hu et al., 2021) LLM to perform a thorough instruction-following tuning.

### 3.4 STAGE 4: PREFERENCE ALIGNMENT WITH DPO

After the instruction tuning phase, the resulting model gains the capability for CoT reasoning on visual math problems. However, it may still produce inaccurate intermediate steps due to insufficient supervision for generating the best reasoning path. To address this, we further apply CoT preference alignment using the DPO (Rafailov et al., 2024) algorithm to further enhance the model's reasoning performance. Specifically, we adopt the instruction-tuned model to first infer CoT reasoning process on the 582K problems generated by data engine within MAVIS-Instruct. Then, we filter out the incorrect outputs (88K data) based on the final answer as the negative reasoning samples in DPO, and directly utilize the annotated CoT process as the positive samples. We only unfreeze the LoRA parameters for DPO training, and finally obtain our mathematical specialist, MAVIS-7B.

## 4 EXPERIMENT

We first detail our experimental settings in Section 4.1, and then discuss the quantitative on different benchmarks and qualitative examples in Sections 4.2 and 4.3, respectively. Please refer to the Appendix for more data details and ablation studies.

Table 2: **Evaluation on Six Mathematical Benchmarks.** 'MMMU-Math' denotes the math problems within the test set of MMMU. 'GPS', 'ALG', and 'GEO' denote geometry problem solving, algebraic, and geometry in MathVista's *testmini* set. 'S1', 'S2', and 'S3' denote different problem steps in We-Math's *testmini* set. '*' denotes previous mathematical visual specialists. The highest scores for closed-source and open-source MLLMs are marked in red and blue, respectively.

| Model | LLM Size | GeoQA | FunctionQA | MMMU-Math | MathVision | MathVista | | | We-Math | | |
|---|---|---|---|---|---|---|---|---|---|---|---|
| | | | | | | GPS | ALG | GEO | S1 | S2 | S3 |
| *Baselines* | | | | | | | | | | | |
| Random Chance | - | 17.1 | - | 21.6 | 7.2 | 24.1 | 25.8 | 22.7 | - | - | - |
| Human | - | 92.3 | - | 84.2 | 68.8 | 48.4 | 50.9 | 51.4 | - | - | - |
| *LLMs* | | | | | | | | | | | |
| ChatGPT | - | - | - | - | 9.7 | 31.7 | 32.4 | 33.0 | - | - | - |
| GPT-4 | - | - | - | 30.6 | 13.1 | 31.7 | 33.5 | 32.2 | - | - | - |
| *Closed-source MLLMs* | | | | | | | | | | | |
| Qwen-VL-Plus | - | - | - | - | 10.7 | 38.5 | 39.1 | 39.3 | - | - | - |
| Qwen-VL-Max | - | - | - | 36.3 | 15.6 | - | - | - | 40.8 | 30.3 | 20.6 |
| GPT-4V | - | - | - | 48.4 | 22.8 | 50.5 | 53.0 | 51.0 | 65.5 | 49.2 | 38.2 |
| *Open-source MLLMs* | | | | | | | | | | | |
| LLaMA-Adapter V2 | 7B | 18.1 | 30.6 | 23.0 | 8.2 | 25.5 | 26.3 | 24.3 | - | - | - |
| mPLUG-Owl2 | 7B | 15.7 | 29.0 | 18.8 | 8.6 | 12.5 | 27.7 | 14.2 | - | - | - |
| UniMath | - | 50.0 | - | - | - | - | - | - | - | - | - |
| LLaVA-1.5 | 13B | 20.3 | 33.9 | 24.0 | 11.2 | 16.3 | 38.5 | 16.7 | - | - | - |
| ShareGPT4V | 13B | - | - | - | 11.9 | - | - | - | - | - | - |
| SPHINX-MoE | 8×7B | - | 33.9 | - | 14.2 | 31.2 | 31.7 | 30.5 | - | - | - |
| G-LLaVA* | 13B | 67.0 | 24.2 | 27.6 | 1.3 | 36.1 | 24.6 | 33.1 | 32.4 | 30.1 | 32.7 |
| Math-LLaVA* | 13B | 62.3 | 38.7 | 36.1 | 15.5 | 57.7 | 53.0 | 56.5 | 37.5 | 30.5 | 32.4 |
| InternLM-XC2. | 7B | 66.4 | 38.7 | 30.1 | 14.5 | 63.0 | 56.6 | 62.3 | 47.0 | 33.1 | 33.0 |
| LLaVA-NeXT | 110B | - | - | - | - | - | - | - | 53.7 | 36.9 | 31.5 |
| **MAVIS-7B w/o DPO*** | 7B | 66.7 | 40.3 | 39.2 | 18.6 | 63.2 | 58.3 | 63.0 | 56.9 | 37.1 | 33.2 |
| **MAVIS-7B*** | 7B | 68.3 | 50.0 | 42.4 | 19.2 | 64.1 | 59.2 | 63.2 | 57.2 | 37.9 | 34.6 |

## 4.1 EXPERIMENTAL SETTINGS

**Implementation Details.** We adopt a CLIP ViT-L (Radford et al., 2021) as the pre-trained model to fine-tune our CLIP-Math, and utilize Mammoth2-7B (Yue et al., 2024) as the base LLM to construct MAVIS-7B. In the first stage, we fine-tune the CLIP for 10 epochs with a batch size 16 and an initial learning rate $2e^{-6}$. In the second stage, we train the diagram-language alignment for 1 epoch with a batch size 32 and an initial learning rate $2e^{-6}$, and adopt LoRA (Hu et al., 2021) with a rank 128. In the third and fourth stages, we adopt the same training settings as the second one.

**Evaluation Schemes.** We evaluate our model MAVIS-7B on several popular mathematical benchmarks, MathVerse (Zhang et al., 2024b), GeoQA (Chen et al., 2021c), FunctionQA (function problems in MathVista (Lu et al., 2023)), MMMU-Math (the math problems in MMMU (Yue et al., 2023a)), MathVision (Wang et al., 2024b), three mathematical categories in MathVista, and We-Math (Qiao et al., 2024). We compare a variety of existing MLLMs, including two mathematical visual specialist (Gao et al., 2023a; Shi et al., 2024), two LLMs (OpenAI, 2023a;b), and other general MLLMs (Bai et al., 2023b; Gao et al., 2023b; Ye et al., 2023b; Liu et al., 2023a; Chen et al., 2023b; Gao et al., 2024; Dong et al., 2024; Liu et al., 2024a; Chen et al., 2023a; Gao et al., 2024).

## 4.2 QUANTITATIVE PERFORMANCE

As shown in Table 1 for the MathVerse benchmark, MAVIS-7B achieves the best overall scores in both CoT evaluation and accuracy among open-source MLLMs with only a 7B model size, and consistently surpasses the second-best method on different problem versions. Specifically, our model surpasses the powerful InternLM-XComposer2 (7B) (Dong et al., 2024) by +9.3% and ShareGPT4V (13B) (Chen et al., 2023b) by +17.8% CoT evaluation scores. Compared to other mathematical visual specialist, i.e., G-LLaVA (7B) (Gao et al., 2023a) and the concurrent Math-LLaVA (13B) (Shi et al., 2024), MAVIS-7B exhibits superior problem-solving capabilities with higher CoT evaluation scores of +19.5% and +11.1%, respectively. In addition, our model is also advantageous to the most powerful open-source MLLM series, LLaVA-NeXT (Li et al., 2024a), from 8B to 110B model sizes, demonstrating the math-specific proficiency of MAVIS-7B. Note that, the improvement brought by DPO (our fourth-stage training) is more apparent in CoT evaluation compared to the accuracy scores, indicating that the preference alignment learning can effectively boost the CoT reasoning capabilities.

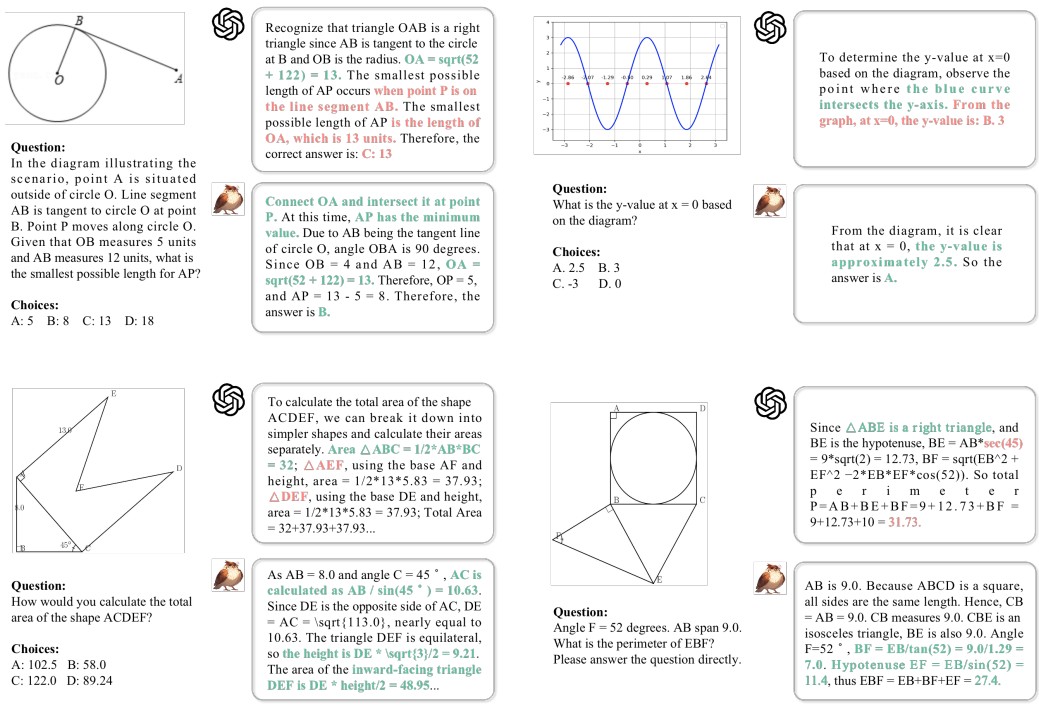

Figure 6: **Problem-solving Comparison of MAVIS-7B and GPT-4V.**

Table 2 showcases the performance comparison on six other mathematical benchmarks, where our model still attains remarkable performance among other MLLMs. In detail, MAVIS-7B outperforms the closed-source Qwen-VL-Max (Bai et al., 2023a) by +6.1% in MMMU-Math, +3.6% in MathVision, and around +10% in three subsets of We-Math. Our model even exceeds GPT-4V (OpenAI, 2023b) in the three mathematical categories of MathVista, indicating our problem-solving and reasoning proficiency. We also observe that, the enhancement from DPO increases from 'S1' to 'S3' of We-Math, which well demonstrates its benefit on math problems with more intricate reasoning steps.

## 4.3 Qualitative Analysis

In Figure 6, we compare the mathematical problem-solving examples between MAVIS-7B and GPT-4V (OpenAI, 2023c). As presented, our model not only showcases better accuracy in understanding the geometric elements, function curves, and coordinate axes in mathematical diagrams, but also performs higher-quality step-by-step reasoning process for formula substitution and numerical calculation. This demonstrates the effectiveness of our four-stage training pipeline and automatic data engine for enhanced diagram understanding and CoT reasoning.

## 5 Conclusion

In this paper, we propose MAVIS, the first mathematical visual instruction tuning paradigm for MLLMs. We first introduce two high-quality datasets by a delicate data engine, MAVIS-Caption and MAVIS-Instruct, containing large-scale diagram-language and problem-solving data. Then, we customize a three-stage training framework to progressively train the math-specific vision encoder, the diagram-language alignment, and the mathematical reasoning capabilities of MLLMs. The obtained specialist model, MAVIS-7B, achieves superior performance across different mathematical visual benchmarks, demonstrating the potential to serve as a new standard for future research.

ACKNOWLEDGEMENT

This project is funded in part by National Key R&D Program of China Project 2022ZD0161100, by the Centre for Perceptual and Interactive Intelligence (CPII) Ltd under the Innovation and Technology Commission (ITC)'s InnoHK, by NSFC-RGC Project N_CUHK498/24. Hongsheng Li is a PI of CPII under the InnoHK.

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

## A APPENDIX

### A.1 RELATED WORK

**Visual Instruction Tuning.** The advancement of large language models (LLMs) (Brown et al., 2020; Jiang et al., 2024; Touvron et al., 2023b; Chiang et al., 2023) with instruction tuning has significantly enhanced zero-shot capabilities across a range of tasks. Drawing inspiration from this, LLaMA-Adapter series (Zhang et al., 2024a; Gao et al., 2023b; Han et al., 2023) propose a zero-initialized attention mechanism to align frozen vision encoders (Radford et al., 2021) with LLaMA (Touvron et al., 2023a) for multi-modal learning. LLaVA series (Liu et al., 2023b;a) employ a linear projector for vision-language alignment, establishing visual instruction tuning as a standard training approach in the multi-modal field. Flamingo (Alayrac et al., 2022) and OpenFlamingo (Awadalla et al., 2023) have honed visual representation by integrating a cross-attention resampler with vision encoders. SPHINX series (Gao et al., 2024; Lin et al., 2023; 2025) and MR-MLLM (Wang et al., 2024a) utilize a blend of visual encoders to make the LLM cognizant of various image aspects. InternVL series (Chen et al., 2024; Dong et al., 2024; Team, 2023) employ a large vision encoder and Q-Former (Li et al., 2022) to incorporate high-quality visual information through a multi-stage training methodology. LLaVA-NexT (Liu et al., 2024a; Li et al., 2024a;b) further introduces the 'AnyRes' technique to manage images at any given resolution, and LLaVA-NexT-Interleave (Li et al., 2024c) extends the scope widely to interleave multi-image settings. There are also recent efforts to apply visual instruction tuning to 3D (Guo et al., 2023; Xu et al., 2023; Guo* et al., 2024; Tang et al., 2025), video (Li et al., 2023a; Fu et al., 2024), reasoning Guo et al. (2025); Jiang et al. (2025); Peng et al. (2024), and robotics Jia et al. (2024); Liu et al. (2024c) scenarios. Despite the impressive strides made in both model capability and training efficiency by multi-modal large language models (MLLMs) through visual instruction tuning, there is currently no MLLM specifically designed for mathematical problem-solving, nor a substantial dataset available for such purposes in the open-source community. In this paper, we mitigate the issue by proposing MAVIS with high-quality mathematical visual datasets and training paradigms.

**Mathematics in Large Models.** Recent research has predominantly concentrated on text-only mathematical problem-solving using LLMs. MAmmoTH (Yue et al., 2023b; 2024) have compiled extensive collections of mathematical problems, training LLMs using the reasoning processes described in solutions. MetaMATH (Yu et al., 2023) has expanded upon this by rewriting existing problems to create a larger dataset. MathCoder (Wang et al., 2024c) and ToRA (Gou et al., 2023) introduced a tools agent approach, employing Python code and symbolic resolvers during the training phase, significantly outperforming traditional models that rely on text-only mathematical reasoning. However, in the multi-modal field, despite the introduction of several datasets such as Geometry3K (Lu et al., 2021), GeoQA (Chen et al., 2021b), UniGeo (Chen et al., 2022), UniMath (Liang et al., 2023), and GeomVerse (Kazemi et al., 2023), aiming at enhancing the performance of MLLMs in solving graphical mathematical problems, these datasets are quite limited in scale and domain. Based on these datasets, G-LLaVA (Gao et al., 2023a) has developed superior capabilities for understanding graphical geometries but struggles with mathematical problems in other domains. The comprehensive benchmark MathVerse (Zhang et al., 2024b) has also highlighted the existing MLLMs' unsatisfactory capacity for encoding visual diagrams in diverse mathematical domains. Therefore, there is a pressing need for the development of more robust encoders for mathematical images and the tuning of MLLMs with mathematical visual instructions, for which we propose MAVIS to address the challenges.

### A.2 HUMAN EVALUATION OF MAVIS-INSTRUCT

To assess the dataset's coverage, validity, and quality, human verification is employed. The creation process of our MAVIS-Instruct dataset can be broadly categorized into two approaches:

- **GPT-generated:** This method leverages GPT-4 to generate new problems (including questions, rationales, and answers) based on existing problems with diagrams. While this approach produces fluent, human-like sentences, it may be influenced by the inherent capabilities and occasional instability of GPT-4V.

- **Data Engine:** As the main source of our mathematical visual data, this method utilizes the custom automatic data engine to generate new problems (including diagrams, questions,

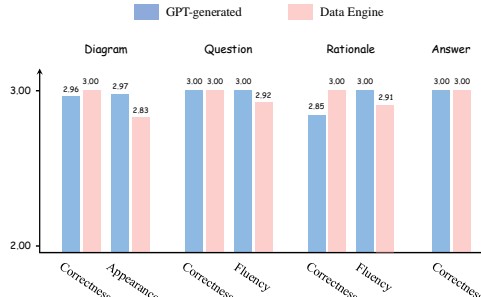
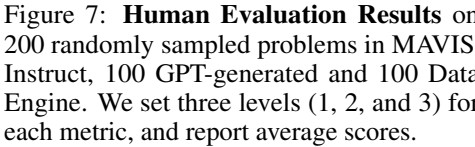
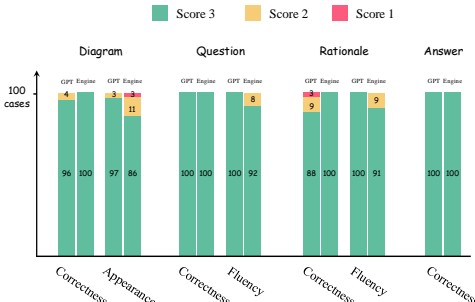

Figure 7: **Human Evaluation Results** on 200 randomly sampled problems in MAVIS-Instruct, 100 GPT-generated and 100 Data Engine. We set three levels (1, 2, and 3) for each metric, and report average scores.

Figure 8: **Human Evaluation Statistics** on 200 randomly sampled problems in MAVIS-Instruct, 100 GPT-generated and 100 Data Engine. We count the numbers of three score levels (1, 2, and 3) for each metric.

rationales, and answers), without relying on GPT models. It guarantees 100% correctness due to the use of rigorous templates, though it may occasionally exhibit rigid expressions.

Specifically, we evaluate four aspects(Diagram, Question, Rationale and Answer) of each problem using seven metrics. Each metric is scored on a scale of 1 to 3, where 1 denotes *poor*, 2 denotes *moderate*, and 3 denotes *good*. The human evaluation results are shown in Figure 7 and score statistics are shown in Figure 8. In addition, we also showcase some specific examples in Figure 9 and Figure 10. We analyze each aspect as follows:

- **Diagram:** The diagrams in GPT-generated problems are directly collected from existing sources with rigorous human filtering, ensuring high quality, resulting in scores close to 3. In contrast, for rule-based problems, the diagrams are drawn accurately using Python code driven by our data engine, which guarantees correctness. However, these diagrams may lack alignment with human aesthetic preferences, as indicated by 3% of them receiving an appearance score of 1.

- **Question:** Regarding the questions, both GPT-generated and rule-based problems display a high degree of accuracy in aligning with the diagram elements. This is attributed to the well-crafted prompts used with GPT-4 and the meticulous template design of the data engine. Nevertheless, rule-based questions may occasionally exhibit minor fluency issues, as they lack human refinement.

- **Rationale:** In terms of the rationales, most instances feature a precise and detailed chain-of-thought (CoT) reasoning process. However, in a few cases (3% receiving an accuracy score of 1), some GPT-generated rationales contain minor reasoning or calculation errors, which are inherent to GPT-4's limitations in problem-solving. These errors usually affect only one or two steps and do not compromise the overall logic. Conversely, the rule-based rationales are highly accurate due to the carefully designed data engine, although there is still room for improvement in language fluency.

- **Answer:** The answers in both methods achieve high correctness scores. For GPT-generated problems, we prompt GPT-4 to identify a known condition from the original problems as the answer. Similarly, for rule-based problems, we randomly select a known attribute from the generated diagrams to serve as the answer.

Overall, the randomly sampled instances show that our dataset exhibits good question quality and answer accuracy.

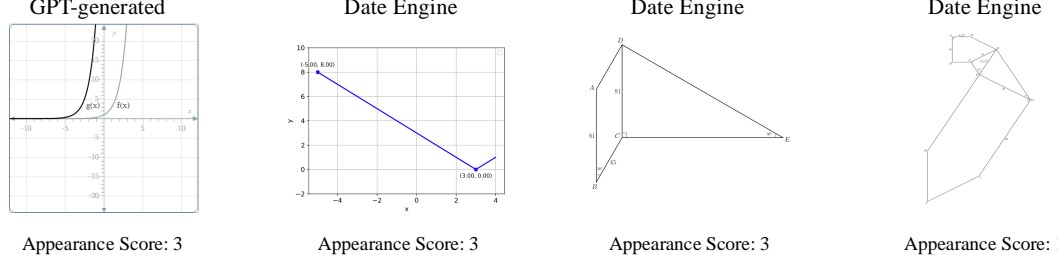

| GPT-generated | Date Engine | Date Engine | Date Engine |
| :---: | :---: | :---: | :---: |
| Appearance Score: 3 | Appearance Score: 3 | Appearance Score: 3 | Appearance Score: 1 |

Figure 9: **Diagram Examples in MAVIS-Instruct.** The first three diagrams showcase superior correctness and appearance, while a small portion of Data Engine generated diagrams (3%) are not aligned with human preference, e.g., the fourth diagram.

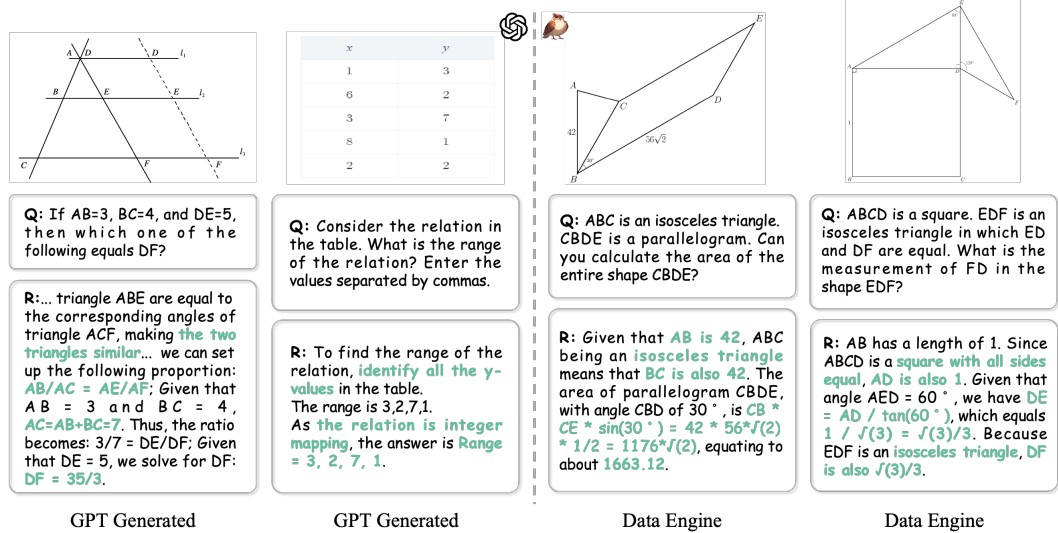

Figure 10: **Accurate Rationale Examples in MAVIS-Instruct.** Most GPT-generated and Data Engine-generated rationales ensure correctness.

Table 3: **Statistics of MAVIS-Caption.**

| Statistic | Number |
|---|---|
| *Total Captions* | |
| - Total number | 588K |
| - Average length (words) | 62.85 |
| - Average length (characters) | 339.68 |
| - Vocabulary size | 418 |
| *Plane Geometry* | |
| - Total number | 299K (50.9%) |
| - Average length (words) | 69.77 |
| - Average length (characters) | 385.85 |
| - Vocabulary size | 195 |
| *Analytic Geometry* | |
| - Total number | 77K (13.1%) |
| - Average length (words) | 39.64 |
| - Average length (characters) | 210.10 |
| - Vocabulary size | 158 |
| *Function* | |
| - Total number | 212K (36.0%) |
| - Average length (words) | 61.48 |
| - Average length (characters) | 321.46 |
| - Vocabulary size | 149 |

Table 4: **Subject Distribution of MAVIS-Instruct.**

| Statistic | Number |
|---|---|
| *Total questions* | 834K |
| - Multiple-choice questions | 615K (62.4%) |
| - Free-form questions | 218K (37.6%) |
| *Data Engine Generated Problems* | 582K |
| - Geometry questions | 466K (80.0%) |
| - Function questions | 116K (20.0%) |
| *Data Engine Captions Annotated by GPT-4* | 51K |
| - Geometry questions | 30K (58.8%) |
| - Function questions | 21K (41.2%) |
| *Manual Collection Augmented by GPT-4* | 83K |
| - Geometry questions | 72K (86.5%) |
| - Function questions | 11K (13.5%) |
| *Existing Datasets Augmented by GPT-4* | 118K |
| - Geometry questions | 118K (100.0%) |
| - Function questions | 0 (0%) |
| Number of unique images | 611K (73.3%) |
| Number of unique questions | 804K (96.5%) |
| Number of unique answers | 675K (81.0%) |
| Average question length | 44.60 |
| Average answer length | 62.82 |

## A.3 ABLATION STUDY

### A.3.1 MAVIS-CAPTION

To validate the enhancement of Math-CLIP's **diagram perception capability**, we sampled 100 validation diagram-caption pairs and computed their cosine similarity using both CLIP and Math-CLIP. The results, as shown in Table 5, indicate that Math-CLIP encodes more discriminative diagram features. Additionally, the attention visualization in Figure 1(a) of the main paper further demonstrates that Math-CLIP captures mathematical visual elements within diagrams more effectively, highlighting the efficacy of MAVIS-Caption.

To validate the role of MAVIS-Caption in second-stage training, we present both quantitative and qualitative results for diagram captioning on the same 100 validation pairs in the first column of Table 6. The use of MAVIS-Caption significantly enhances the **diagram understanding capability**. This shows that MAVIS-Caption helps the LLM generate accurate captions from diagrams, improving its ability to comprehend each visual token from Math-CLIP and align visual elements with textual descriptions. We also evaluated MAVIS's performance on MathVerse without second-stage training, as shown in the second column of Table 6. Without MAVIS-Caption training, the CoT reasoning quality of MAVIS-7B is somewhat compromised. This suggests that training the model in diagram captioning improves its **mathematical expression capability**, enabling it to produce language expressions that align with mathematical concepts. This foundational skill supports the generation of subsequent CoT reasoning steps.

Table 5: **Diagram Perception Enhancement by Math-CLIP**, using MAVIS-Caption in the first stage. We calculate the average cosine similarity among 100 validation diagram-caption pairs.

| Vision Encoder | Matched Pair ↑ | Unmatched Pair ↓ |
|---|---|---|
| CLIP | 0.22 | 0.24 |
| **Math-CLIP** | **0.83** | **0.17** |

### A.3.2 MAVIS-INSTRUCT

**Redundant Text** When curating questions for MAVIS-Instruct, we minimize the redundant content within the question texts, which refers to the directly observable content in the diagram, e.g., the

Table 6: **Diagram Understanding Enhancement** and **Mathematical Expression Enhancement** in LLM using MAVIS-Caption in the second Stage. We compare the METEOR and CIDEr scores for diagram captioning on 100 validation samples, as well as the accuracy and CoT evaluation results on MathVerse, both with and without the MAVIS-Caption training.

| Training Data | Diagram-Caption Pairs | | MathVerse | |
|---|---|---|---|---|
| | METEOR | CIDEr | Acc (%) | CoT-E (%) |
| **w MAVIS-Caption** | 23.7 | 161.3 | 28.4 | 35.2 |
| w/o MAVIS-Caption | 14.0 | 69.4 | 25.6 | 32.8 |

presence of shapes or intersection points of functions. Such information is repetitive to visual components, and may assist MLLMs in bypassing the process of diagram interpretation, thereby harming their related skills. By mostly avoiding redundant texts in MAVIS-Instruct, our data enforces MLLMs to learn stronger **diagram interpretation capabilities**. In Table 7, we add redundant texts (diagram captions) to the Data Engine Generated Problems for training, leading to expected performance drop.

**CoT Rationales**    For each instance in MAVIS-Instruct, we incorporate detailed rationales for problem-solving, either generated by GPT-4 or our rule-based data engine. In Table 8, we remove all intermediate rationales of each problem in MAVIS-Instruct, and train the model to directly output the final answer. As shown, both the CoT evaluation and accuracy scores are degraded. This demonstrates the significance of our rationale annotations, which effectively improves the CoT **reasoning capabilities** of MLLMs.

Table 7: **Diagram Interpretation Enhancement for MLLM**, using MAVIS-Instruct in the third stage. We compare the results by adding redundant texts (diagram captions) to the Data Engine Generated Problems within MAVIS-Instruct.

| MAVIS-Instruct | MathVerse | GeoQA | FunctionQA |
|---|---|---|---|
| **w/o Redundant Texts** | **28.4** | **68.3** | **50.0** |
| w Redundant Texts | 26.5 | 66.5 | 48.4 |

Table 8: **Reasoning Capability Enhancement for MLLM**, using MAVIS-Instruct in the third stage.

| Training Data | MathVerse | |
|---|---|---|
| | Acc | CoT-E |
| **w Rationales** | **28.4** | **35.2** |
| w/o Rationales | 25.2 | 26.6 |

### A.3.3 COMPARED TO GENERAL VISUAL INSTRUCTION DATA

Since Mammoth-2 is a highly capable LLM for mathematical tasks, one possible question is whether simply integrating a vision encoder into Mammoth-2 and training it with conventional visual instruction tuning data would suffice for effectively solving visual-based mathematical problems. To compare MAVIS data with other visual instruction tuning datasets and investigate the specific benefits of MAVIS data in Mammoth-2 (7B), we conduct an ablation study. We utilize the data from LLaVA-NeXT (558K for pre-training and 760K for fine-tuning) and compare it with our MAVIS data (558K MAVIS-Caption for pre-training and 834K MAVIS-Instruct for fine-tuning). Performance is evaluated using the accuracy metric on MathVerse, excluding the DPO training stage for fairness.

Table 9: Ablation study results for comparison between MAVIS Data and other visual instruction tuning data. The first row in the table represents the original LLaVA-NeXT-8B.

| Visual Encoder | LLM | Pre-training | Fine-tuning | MathVerse Acc (%) |
|---|---|---|---|---|
| CLIP | LLaMA-3 (8B) | LLaVA data | LLaVA data | 15.6 |
| CLIP | Mammoth-2 (7B) | LLaVA data | LLaVA data | 18.3 |
| CLIP | Mammoth-2 (7B) | LLaVA data | **MAVIS-Instruct** | 25.7 |
| CLIP | Mammoth-2 (7B) | **MAVIS-Caption** | **MAVIS-Instruct** | 26.4 |
| **Math-CLIP** | Mammoth-2 (7B) | **MAVIS-Caption** | **MAVIS-Instruct** | 27.5 |

Based on the results presented in Table 9, we make the following observations:

1. **Mammoth-2 vs. LLaMA-3:** Mammoth-2 achieves a +2.7 improvement in accuracy compared to LLaMA-3, highlighting its prior knowledge and inherent capability in mathematical problem solving.

2. **Impact of MAVIS-Instruct:** Fine-tuning with MAVIS-Instruct significantly enhances performance by +7.4, underscoring the substantial advantage of our dataset for mathematical reasoning tasks compared to general visual instruction datasets.

3. **MAVIS-Caption and Math-CLIP:** Using MAVIS-Caption for pre-training and employing the Math-CLIP encoder further boosts performance, leading to enhanced mathematical visual perception and reasoning capabilities. Overall, our MAVIS data contributes a +9.2 improvement in accuracy over Mammoth-2 trained with LLaVA data.

### A.3.4 PERFORMANCE ACROSS DIFFERENT SUBJECTS

Although MAVIS-Instruct contains a substantial number of high-quality solid geometry problems that were manually curated, our data engine only generates plane geometry and function problems. Therefore, we aim to evaluate the performance of the MAVIS model across different mathematical domains, specifically plane geometry, functions, and solid geometry. We provide the detailed subject scores of MAVIS-7B on MathVerse, comparing the CoT evaluation score (note that the subject-level accuracy scores are not publicly released) with other models on the official leaderboard.

Table 10: Performance comparison across different models on Plane Geometry, Solid Geometry, and Functions of MathVerse evaluation tasks.

| Model | All (CoT-Eval) | Plane Geometry | Solid Geometry | Functions |
|---|---|---|---|---|
| LLaVA-NeXT | 17.2 | 15.9 | 19.6 | 23.1 |
| ShareGPT4V | 17.4 | 16.9 | 15.0 | 20.2 |
| SPHINX-MoE | 22.8 | 24.5 | 15.8 | 19.5 |
| InternLM-XC2 | 25.9 | 26.2 | 20.1 | 23.7 |
| **MAVIS-7B** | **35.2** | **37.1** | **28.9** | **31.0** |

The results shown in Table 10 demonstrate that our model achieves leading performance across all three subjects. Notably, its proficiency in plane geometry and functions can be attributed to the training with our meticulously curated MAVIS dataset. Additionally, for solid geometry, which shares similarities with plane geometry in both visual appearance and reasoning process, we believe that our model effectively generalizes its learned knowledge and reasoning capabilities, leading to enhanced performance in this domain as well.

### A.3.5 SYNTHETIC DATA VS REAL DATA

In MAVIS-Instruct, we integrate both synthetic problems generated by the data engine (633K, 76%) and real-world problems augmented with GPT (201K, 24%). The synthetic data is composed of both geometry and function problems, while the real-world data primarily focuses on geometry. We conduct an ablation study to assess the contributions of these data components, excluding the DPO training stage to ensure fairness.

Table 11: Ablation study of synthetic and real data contributions to MAVIS-7B's performance.

| Synthetic Data | Real-world Data | MathVerse Acc (%) | GeoQA | FunctionQA | MMMU-Math |
|---|---|---|---|---|---|
| ✓ | – | 22.6 | 44.2 | 37.1 | 34.6 |
| – | ✓ | 24.3 | 66.4 | 25.8 | 29.8 |
| ✓ | ✓ | 27.5 | 66.7 | 40.3 | 39.2 |

The results shown in Table 11 indicate that the two data sources exhibit complementary characteristics, both playing a crucial role in achieving the final performance. Specifically, synthetic data significantly enhances the results on FunctionQA and MMMU-Math, as these benchmarks include a substantial

proportion of function-related problems. Conversely, real-world data has a greater impact on GeoQA, given its stronger alignment with the geometry-focused nature of this benchmark.

### A.3.6 DATA SCALING

A good instruction tuning dataset should exhibit the characteristic of data scaling: as the dataset size increases, the model trained on it should demonstrate progressively better performance. To verify that MAVIS-Instruct possesses this property, we conduct an ablation study on the 834K MAVIS-Instruct dataset by randomly sampling 25%, 50%, and 75% of the data for instruction tuning, excluding the DPO stage. We then evaluate the models using the accuracy metric on MathVerse. The results, as shown in Table 12, indicate that the performance of MAVIS-7B consistently improves as the data scale increases. This demonstrates the promising potential of our dataset to further enhance mathematical reasoning capabilities with larger-scale utilization.

Table 12: Performance of MAVIS-7B at different data proportions.

| 25% | 50% | 75% | 100% |
|------|------|------|------|
| 23.3 | 25.7 | 26.9 | 27.5 |

Table 13: Comparison of different training settings.

| LLMs | Caption CIDEr | MathVerse Acc (%) |
|------|------|------|
| Frozen | 79.6 | 26.2 |
| Unfrozen | 146.2 | 28.1 |
| LoRA-based | 161.3 | 28.4 |

### A.3.7 GENERALIZATION ABILITY

Although our Data Engine considers as many problem types as possible, it is inherently challenging for a manually designed engine to cover all possible types of mathematical visual diagrams. To alleviate this, we incorporate real-world data by manual collection and GPT augmentation, which well enhances the generalization capablitity of MAVIS-7B. In Figure 11, we selected examples involving inscribed triangles (they cannot be generated via our data engine) from publicly available math benchmarks, and tested MAVIS-7B on these examples, where our model performs accurate reasoning.

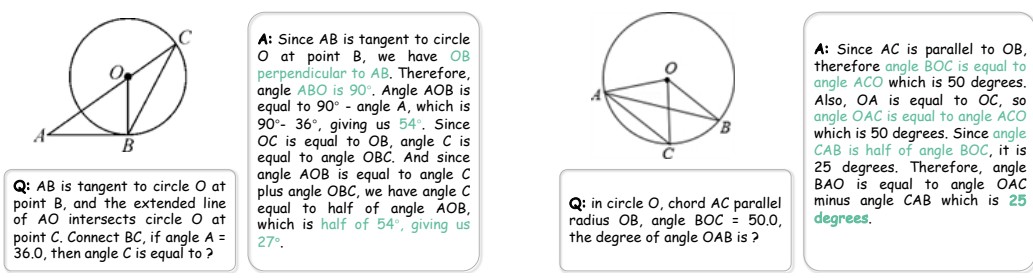

Figure 11: Examples for MAVIS-7B solving math problems with circumcircles of triangles.

### A.3.8 ENHANCING VISION-ONLY PERFORMANCE

To evaluate the impact of OCR datasets on MAVIS-7B's performance, we conducted an ablation study focusing on vision-only and vision-dominant problems in MathVerse. These problems require the model to interpret question texts rendered directly in diagrams, thus relying heavily on OCR capabilities. MAVIS-7B, however, was initially not trained with OCR-specific datasets, limiting its performance in these tasks.

In contrast, generalist models like LLaVA-NeXT include extensive OCR datasets such as OCRVQA, DocVQA, and SynDog-EN, which significantly enhance their OCR capabilities. To bridge this gap, we incorporated OCR datasets (OCRVQA and DocVQA) in our third-stage instruction tuning to improve MAVIS-7B's OCR performance.

The results, as shown in Table 14, indicate a notable improvement in vision-dominant and vision-only problems for MAVIS-7B after the inclusion of OCR datasets, highlighting the potential of better OCR

integration for further boosting its performance. In Figure 12, we also showcase some failure cases of our MAVIS-7B with OCR training on vision-only problems. Although the vision-only results are improved via the OCR instruction dataset, the model still suffers from limited perception capabilities of questions and visual elements within the diagram. This indicates that the OCR capability is still the bottleneck of vision-only performance. We leave this as a future work to further enhance the OCR capabilities of MAVIS for mathematical visual elements.

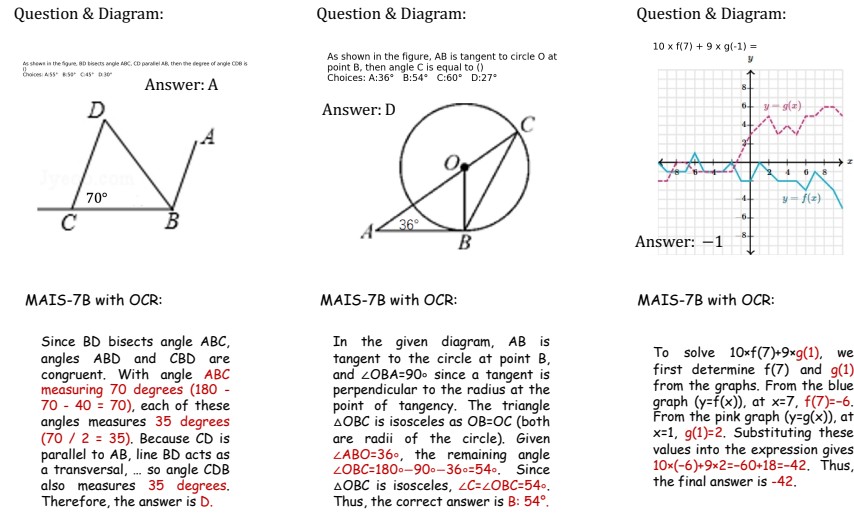

Figure 12: Failure cases of MAVIS-7B with OCR training on vision-only problems.

Table 14: Impact of OCR data on solving vision-only problems.

| Model | LLM Size | All | Text Dominant | Text Lite | Vision Intensive | Vision Dominant | Vision Only |
|---|---|---|---|---|---|---|---|
| LLaVA-NeXT | 8B | 15.6 | 19.4 | 15.2 | 16.8 | 15.2 | 11.3 |
| LLaVA-NeXT | 110B | 24.5 | 31.7 | 24.1 | 21.0 | 22.1 | 20.7 |
| MAVIS-7B | 7B | 28.4 | **41.6** | **29.5** | **27.9** | 24.7 | 18.3 |
| MAVIS-7B w/ OCR | 7B | **28.9** | 40.8 | 29.2 | 27.4 | **26.2** | **21.1** |

### A.3.9 BASE LLM

We investigate different LLMs for the MAVIS model. As shown in Table 15, MAVIS is not very sensitive to LLM choices, and still surpasses previous models with the same LLM.

Table 15: **Performance Comparison using Different LLMs.** We compare the accuracy and CoT evaluation results on MathVerse.

| Method | Base LLM | MathVerse | |
|---|---|---|---|
| | | Acc | CoT-E |
| SPHINX-Plus | LLaMA2-13B | 12.2 | 14.0 |
| ShareGPT4V | Vicuna-13B | 13.1 | 17.4 |
| InternLM-XC2. | InternLM2-7B | 16.5 | 25.9 |
| **MAVIS** | LLaMA2-13B | 24.5 | 30.7 |
| | Vicuna-13B | 24.8 | 30.6 |
| | InternLM2-7B | 28.0 | 33.8 |
| | **MAmmoTH2** | **28.4** | **35.2** |

### A.3.10 DIFFERENT TRAINING SETTINGS

Our training strategy is similar to LLaVA, but with key differences in the pre-training stage: we train both the projection layer and the LoRA-based LLM, whereas LLaVA only trains the projection

layer. This design choice stems from the fundamental differences between general visual tasks and mathematical tasks:

1. For general visual tasks (e.g., LLaVA), training MLLMs typically requires the LLM to generate daily natural language responses, such as descriptive captions or instruction-following outputs. These outputs often rely on pre-existing knowledge within the pre-trained LLM. As a result, in LLaVA, there is no need to unfreeze the LLM to learn new types of outputs.

2. In contrast, for mathematical domains, LLMs need to generate math-specific responses, such as geometric descriptions, functional explanations, formulas, and theorems. These outputs often involve domain-specific knowledge not inherent in pre-trained LLMs. Given this, we incorporate learnable LoRA layers to infuse new knowledge into the LLM, enhancing its capability to produce high-quality mathematical expressions. Concurrently, we aim to prevent the LLM from overfitting to diagram captioning tasks during alignment. Therefore, using LoRA-based tuning allows us to preserve the LLM's generalizable pre-trained language knowledge while injecting specialized math-specific capabilities.

To further investigate the impact of different training settings on model performance, we conduct an ablation study comparing various LLM training settings during the alignment stage. We evaluate two tasks: the CIDEr score for diagram captioning on 100 validation samples (following the same setting as in Table 6 of the Appendix) and the accuracy score on MathVerse. The results, as shown in Table 13, indicate that the LoRA-based approach performs best, enabling MLLMs to generate high-quality mathematical captions while preserving pre-trained knowledge for improved problem-solving capabilities.

### A.3.11  Enhancing a Pre-trained MLLM

To investigate whether our curated data and training techniques can improve the mathematical performance of a pre-trained large model (LLaVA-NeXT), we conducted an ablation study. Specifically, we progressively employed MAVIS-Instruct for instruction tuning, followed by DPO alignment on top of LLaVA-NeXT-8B, with both training stages performed for one epoch using a learning rate of $1 \times 10^{-5}$. The results, as shown in Table 16, demonstrate that these two continual training stages significantly enhance LLaVA-NeXT's ability to solve mathematical problems, with notable improvements across all evaluation categories.

Table 16: Performance improvement of LLaVA-NeXT-8B with MAVIS-Instruct and DPO alignment.

| Model | LLM Size | All | Text Dominant | Text Lite | Vision Intensive | Vision Dominant | Vision Only |
|---|---|---|---|---|---|---|---|
| LLaVA-NeXT | 8B | 15.6 | 19.4 | 15.2 | 16.8 | 15.2 | 11.3 |
| + MAVIS-Instruct | 8B | 22.8 | 32.3 | 25.3 | 24.6 | 18.3 | 14.2 |
| + DPO | 8B | **24.0** | **33.7** | **26.9** | **25.4** | **19.1** | **15.1** |

### A.4  Details of Automatic Data Engine

### A.4.1  Diagram Generation

In this section, we detail the implementation specifics of the process for generating diagrams related to plane geometry, analytic geometry, and function domains.

**Plane Geometry Diagram.**  Inspired by previous multi-hop reasoning methods (Kazemi et al., 2023; Wei et al., 2022; Nye et al., 2021), we employ an iterative generation method over logical theories to generate plane geometric images along with corresponding captions and question-answering pairs, whose complexity can be controlled across multiple axes. Specifically, we first define a set of fundamental geometric shapes in Figure 13.

Within each shape, new basic shapes can be generated by extending a particular edge. For each basic shape, we initially define a meta reasoning process:

$$O_{n-1}, C^i_{m_{n-1}} \xrightarrow{\mathrm{E}^i_{m_{n-1}}} O_n, i \in [1, z], \tag{1}$$

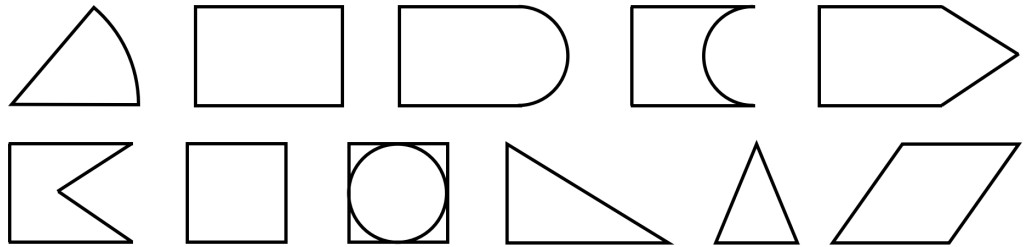

Figure 13: The set of fundamental shapes in plane geometry diagrams, whose straight edges can be extended into other basic shapes.

where $O$ represents the initial side length of the shape, $C_m$ denotes the additional conditions required to complete meta reasoning, and $E_m$ provides a detailed explanation of the meta reasoning process. For example, when considering an isosceles triangle as the $(n-1)^{th}$ shape in a sequence, the vertex angle is still required as $C_m$ to reason about base side length, and then to expand to the $n^{th}$ shape, with $E_m$ serving as the explanation of this process. The variable $z$ indicates that there are $z$ sets of possible meta reasoning for the shape, $n$ indicates the length of the generating sequence, which is also the number of hops of reasoning required to answer the question. The initial side, extend side, and additional conditions for meta-reasoning of each basic shape can be referred to in Figure 13. In the final shape, question-answering pairs pertinent to this shape can be generated as

$$O_n, C_{q_n}^j, Q_n^j \xrightarrow{\text{E}_{q_n}^j} A_n^j, j \in [1, m], \tag{2}$$

where $C_q$ represents the additional conditions required to solve the problem, while $Q$ and $A$ denote the question and answer, respectively. $E_q$ refers to the detailed explanation of the solving process. The variable $m$ indicates that there are $m$ pairs of question-answering and corresponding detailed explanations within the shape. By applying meta reasoning to the $n-1$th shape, the initial side length of the $n$th shape can be deduced. Therefore, for a complex composite figure consisting of $n$ shapes, the overall question-answering pair can be defined as follows:

$$O_1, \sum_{k=1}^{n-1} C_{m_k}, C_{q_n}^j, Q_n^j \xrightarrow{\text{E}_{q_n}^j} A_n^j. \tag{3}$$

Each shape defines a sufficient number of conditions, explanations, and answers to ensure the diversity of the generated question-answering pairs. Based on the aforementioned rules, controlling the length of the generation sequence can regulate the number of reasoning steps, and controlling the type of questions can manage the knowledge required for solving the problems. Thus, we can generate questions of varying difficulty levels, which can also be illustrated in Figure 14a.

**Analytic Geometry Diagram.** The image generation method for analytic geometry is relatively straightforward. First, we randomly select a range within the coordinate system: the minimum value of $x$ is chosen as an integer between $[-12, -8]$, and the maximum value of $x$ is chosen as an integer between $[8, 12]$; the range for $y$ is the same as for $x$. Then, we define the following basic shapes: point, line segment, line, circle, ellipse, rectangle, square, polygon, and sector. During the generation process, we select a number between 1 and 4 as the number of shapes to generate. The generation rule is that **nonlinear shapes** other than points, line segments, and lines **must not overlap**.

**Function Diagram.** The generation of function graphs is also straightforward as shown in Figure 14b. We define the following basic functions, each with a set of parameters that can be randomly selected:

| | |
|---|---|
| **Sine Function** | $y = A \cdot \sin(f \cdot x + \phi)$, where the amplitude $A$ is a random integer between 1 and 3, the frequency $f$ is either 1 or 2, and the phase $\phi$ is a random integer between 0 and $2\pi$. |
| **Cosine Function** | $y = A \cdot \cos(f \cdot x + \phi)$, where the amplitude $A$ is a random integer between 1 and 3, the frequency $f$ is either 1 or 2, and the phase $\phi$ is a random integer between 0 and $2\pi$. |

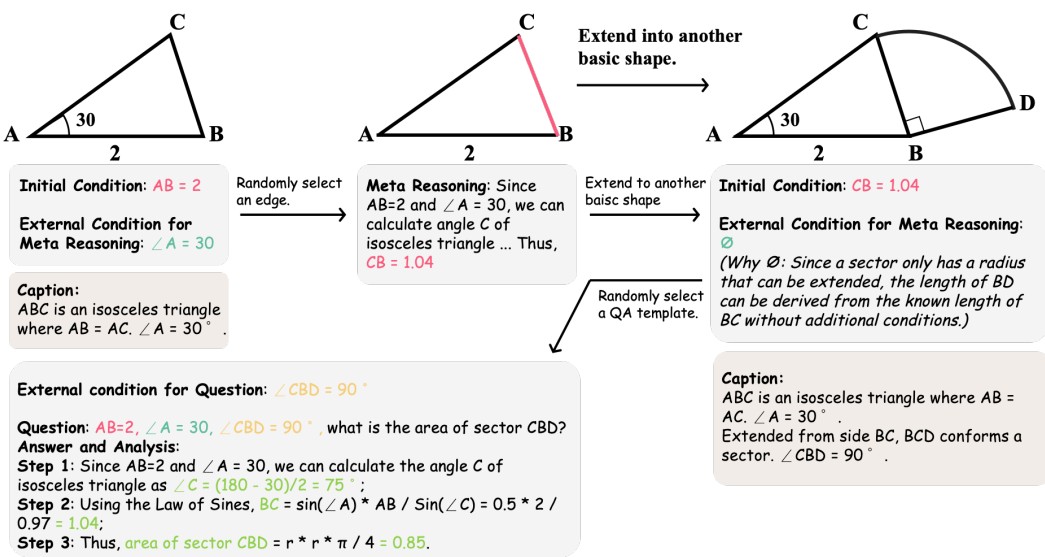

(a) A single process for generating plane geometry diagrams and corresponding question-answering pairs as well as image captions. In this example, the generation sequence length is specified as 2. Initial side length is painted in pink, $C_m$ is painted in green, while $C_q$ is painted in yellow. Whenever a new basic shape is generated, its caption is appended to the previous caption.

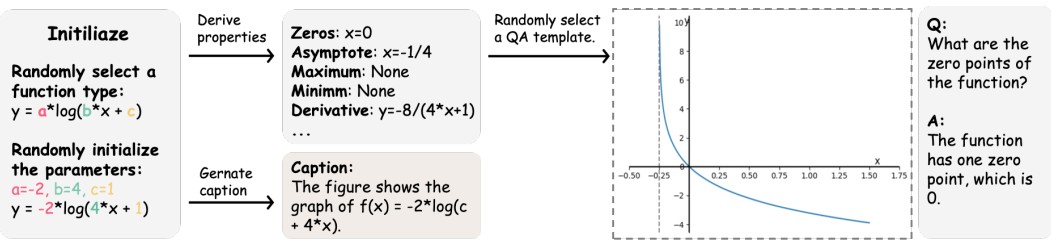

(b) A single process is used for generating function diagrams along with the corresponding question-answer pairs and image captions. Once the functional expression is determined, all its properties can be directly computed, and the function plot can be generated accordingly. The caption for the function diagram simply states the functional expression.

Figure 14: The pipeline of our data engine, consisting of (a) the generation of plane geometry diagrams and (b) the generation of function diagrams.

| | |
|---|---|
| **Tangent Function** | $y = A \cdot \tan(f \cdot x + \phi)$, where the amplitude $A$ is a random integer between 1 and 3, the frequency $f$ is either 1 or 2, and the phase $\phi$ is a random integer between 0 and $2\pi$. |
| **Polynomial Function** | $P(x) = a_n x^n + a_{n-1} x^{n-1} + \cdots + a_1 x + a_0$, where the degree $n$ is a random integer between 1 and 4. The coefficients $a_i$ are randomly selected integers ranging from -3 to 3. |
| **piece-wise Function** | piece-wise polynomial functions are divided into 2 or 3 segments, with each segment's parameters identical to those of a polynomial function. |
| **Logarithmic Function** | $y = a \cdot \log_b(c \cdot x + d)$, where the coefficient $a$ is randomly chosen from $\{-3, -2, -1, 1, 2, 3\}$, the base $b$ is randomly chosen from $\{2, 10, \lfloor e \rfloor\}$, the coefficient $c$ is a random integer between 1 and 3, and the coefficient $d$ is a random integer between 1 and 6, ensuring that $c \cdot x + d$ is positive. |
| **Absolute Function** | $y = |a \cdot x + b|$, where a and b are random integer between $-5$ and 5. |

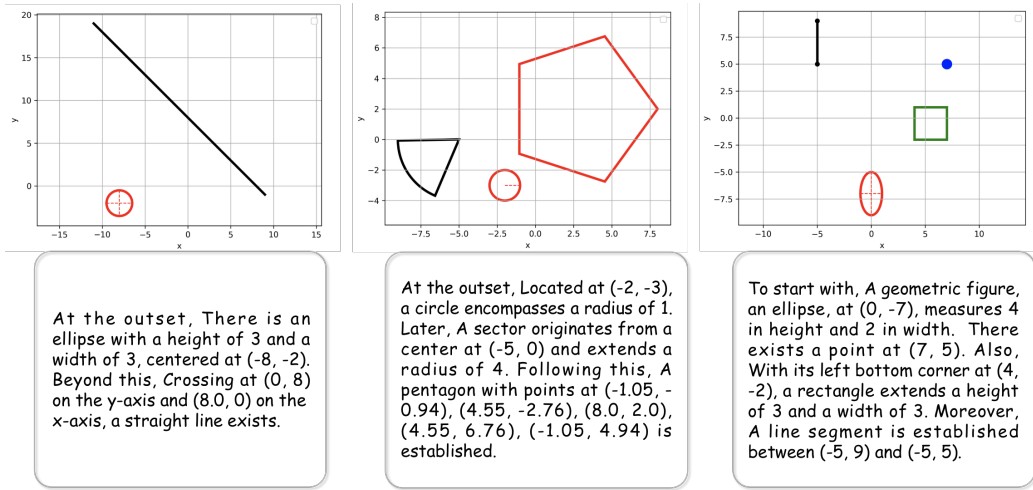

Figure 15: Examples of analytical geometry diagram caption.

We first determine the domain range to be displayed on the function graph. For trigonometric functions, the domain is set to $[-\pi, \pi]$. For piece-wise polynomial functions, the minimum value of $x$ is a random integer between $[-12, -8]$, and the maximum value of $x$ is a random integer between $[8, 12]$. For other functions, the minimum and maximum values of $x$ are random integers within the ranges of $[-6, -3]$ and $[3, 6]$, respectively. During the plotting process, we calculate the local maxima, minima, and zeros of the function by iterating through the domain. We then render the x-coordinates of these extrema and zeros on the x-axis of the function graph.

### A.4.2 MAVIS-CAPTION

In this section, we detail how the captions corresponding to images in the MAVIS-Caption Dataset are generated with our automatic data engine.

**Plane Geometry Caption.** Based on the generation process described in Section A.4.1, when generating each shape, a caption is randomly selected from a set of captions for that shape and some connecting words are randomly added. We also randomly select some edges or angles and state their measurements in the caption. After generating the raw caption, we use GPT-3.5 to refine it, enhancing its linguistic structure and semantic diversity. An example is shown in Figure **??**.

**Function Caption.** According to the function graph generation process described in Section A.4.1, we record the function's zeros and extrema. Additionally, we also record the function's expression and asymptotes. These attributes are incorporated into a randomly selected caption template to form the function graph's caption. Some examples are provided in Figure 16.

**Analytic Geometry Caption.** For each shape, we maintain a set of caption templates that describe the shape's type, coordinate position, and other attributes. In the generation process described in Section A.4.1, we select a template and randomly add some diverse connecting words to form a complete caption. Examples of some captions are shown in Figure 15.

### A.4.3 MAVIS-INSTRUCT

**Manual Collection Augmented by GPT-4.** To complement the dataset with real-world problem-solving scenarios, we hire 8 human experts to manually collect visual math problems from various public sources[1,2,3], spanning plane geometry, analytic geometry, and function. For problems, we try to obtain their content as complete as possible, including questions, diagrams, answers, and rationales if available. The collection process consists of the following steps:

1. **Problem Collection:** We gathered problems from three public sources as comprehensively as possible, including questions, diagrams, answers, category information, and rationales where

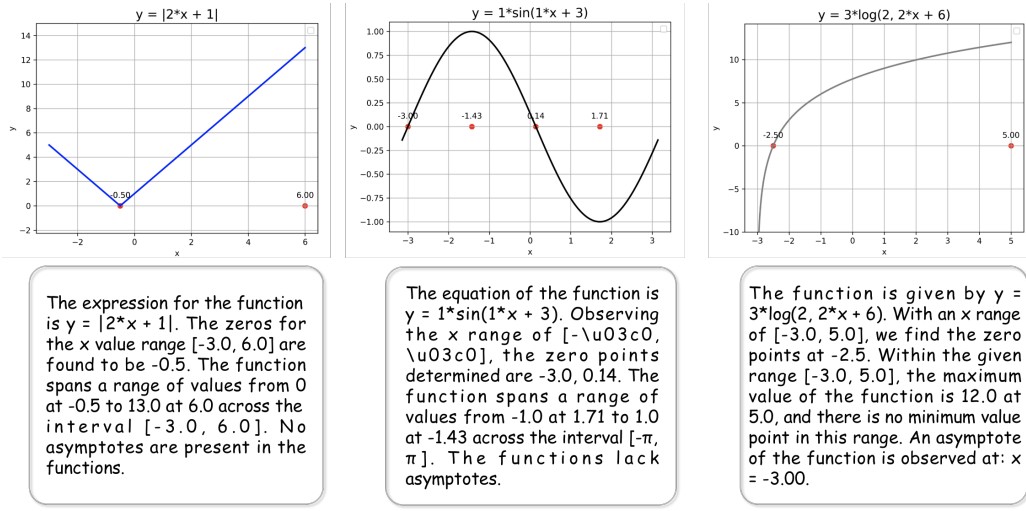

Figure 16: Function diagram captions.

available. The problems are primarily at the high-school level, covering plane geometry and functions (including analytic geometry).

2. **Data Verification:** Based on their initial categories (subject, subfield, and difficulty level), the problems were organized into distinct groups. Six expert annotators were tasked with meticulously verifying the correctness and completeness of each problem. They refined the detailed chain-of-thought (CoT) rationales and *ensured that there was no overlap with evaluation data by visually inspecting the diagrams*. This rigorous verification process resulted in a total of 4K verified problems.

3. **Text-lite Construction:** To optimize the problems for training mathematical visual capabilities, the 4K problems were processed using GPT-4V with a customized prompt (as shown in Figure 15). This step involved removing redundant information from the question text to create concise, text-lite problems, specifically tailored to our training objectives.

Then, we first feed all the related information into GPT-4V to eliminate the redundant information within text questions, constructing the text-lite version of problems by the prompt in Figure 17. Then, we design three types of prompts for GPT-4 to augment 15 multiple-choice questions (including 10 multiple-choice and 5 binary-choice, i.e., 'True' or 'False') and 5 free-form questions, respectively, as shown in Figure 18. We do not adopt GPT-4V here, since GPT-4V itself would misunderstand diagrams for low-quality data augmentation. The newly generated problems contain detailed CoT rationales and diverse question forms.

**Existing Datasets Augmented by GPT-4.** Previous efforts have been made to provide some small-scale, plane geometry datasets, e.g., GeoQA (Chen et al., 2021c), GeoQA+ (Chen et al., 2021a), and Geometry3K (Lu et al., 2021). Although they are limited in data scale for tuning MLLMs and include no rationales, we can also regard them as a seed dataset and adopt GPT-4 to augment larger-scale training data. We do not utilize GPT-4V here for the same reason aforementioned. In detail, we design 3 types of question generation approaches using different prompts, as shown in Figure 19. For Geometry3K, as the question texts are normally brief and contain marginal descriptive information, posing challenges for GPT-4 to understand the diagram, we only augment them to generate binary-choice questions, i.e., 'Ture' or 'False'. For GeoQA+, we can leverage the sufficient redundant information within their texts to generate more diverse and accurate multi-choice and free-form questions. Likewise, GPT-4 can produce CoT rationales for each problem.

---

[1] https://homework.study.com
[2] https://www.ixl.com/math
[3] https://mathspace.co/us

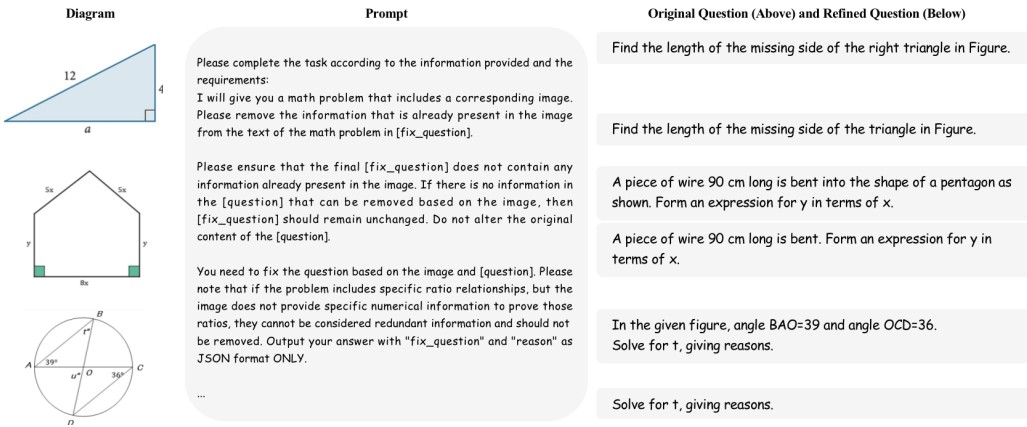

Figure 17: Manually collect visual math problems text-lite version.

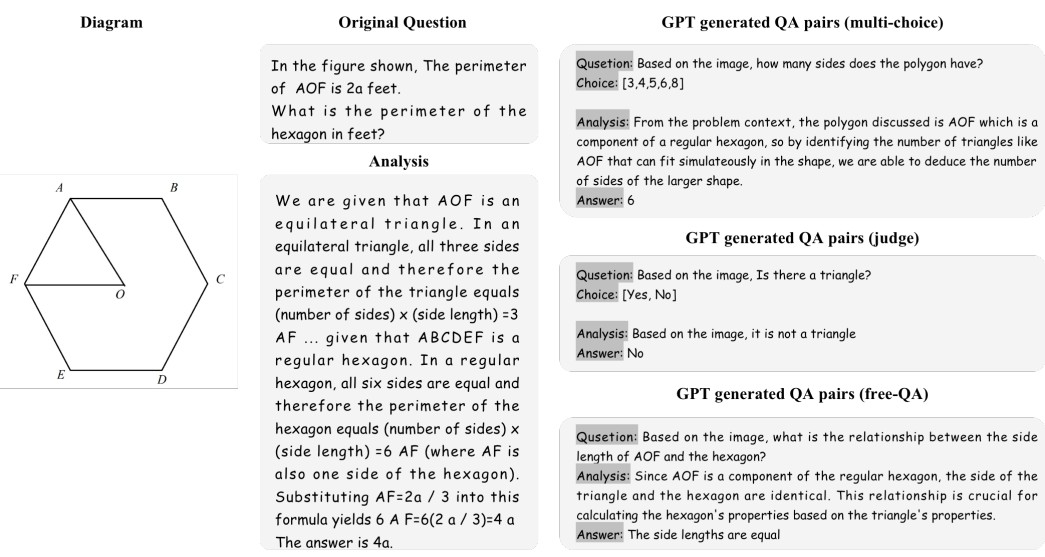

Figure 18: We design different types of prompts for GPT-4 to augment 15 multiple-choice questions and 5 free-form questions, respectively.

**Data Engine Captions Annotated by GPT-4.** Given the delicately designed data engine for automatic diagram-caption creation, we can utilize the generated large-scale pairs to annotate question-answering data using GPT-4V. Different from the previous two sources that augment questions based on questions, we utilize the GPT-4V model here for caution data with two reasons: first, the detailed caption from our data engine can well guide GPT-4V for relatively higher-quality visual embedding; second, the visual input serves as guidance to provide additional spatial information for broad question forms. As shown in Figure 27 and Figure 28, we adopt different prompts for function and plane geometry problems, ensuring that the generated question-answering data is of high quality for instruction tuning.

**Data Engine Generated Problems:**

PLANE GEOMETRY. Based on the generation process described in Section A.4.1, we pose questions about the final shape in the generation sequence. We designed 6 types of questions: finding the perimeter, finding the area, finding the base length, finding the angle, finding the arc length, and finding the extended edge length. Each type of question has a set of templates that can be randomly

**Prompt**

Please modify and amplify the provided information according to the following principles

The content you output should follow the following principles:
"Question": A question that needs to be modified based on the information provided, you can Modify the numerical values, Reask questions based on "Answer" and "Analysis", Represent the question using a different representation method, etc.
"Choice": Based on the options provided by the Question. Choice can be modified accordingly. If the given question does not require a choice, then the choice field is: null. Choices should be as diverse as possible, and you can provide some irrelevant options as interference. The JSON you provided should include an example with a null Choice
"Analysis": Detailed analysis and answers to your "Question"
"Answer": The answer to this question. If Choice is not null, it needs to correspond to the content in Choice. If Choice is null, only output "Analysis" to obtain the final answer.
......

**GPT generated enhence data**

Question: Given Point P (-4, -5), Point Q (-3,-3), Point R (-4,-1), and Point S (-5, -3); Are QR and PS parallel?
Choice: [Yes, No]

Analysis: For lines to be parallel they must have the same gradient.
Answer: Yes

Qusetion: For Line IJ, with I (0, 0) and J (1, 2), and Line KL, with K (0, 5) and L (-1, 3), How are their slopes calculated separately?
Choice: null

Analysis: Equal slopes imply parallel lines. The calculation of slope and the correlation between coordinates x and y
Answer: Slope of IJ = (y2 - y1) / (x2 - x1), Slope of KL = (y4 - y3) / (x4 - x3)

Question: Given Point P (-1, -1), Point Q (0,1), Point R (-1,6) and Point S (-2, 4); What type of shape does PQRS consist?
Choice: A. Parallelogram B. Rentangle C. Sector D. Triangle

Analysis: Oppsite edge are parallel, So PQRS is a parallelogram.
Answer: A. Parallelogram

Figure 19: We design 3 types of question generation approaches using different prompts to augment existing visual mathematical dataset.

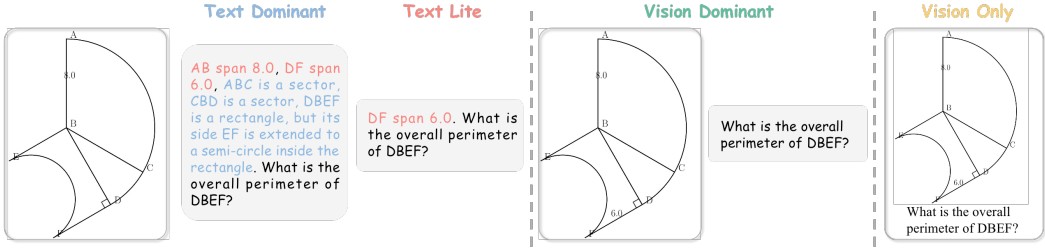

Figure 20: The Text Dominant, Text Lite, Vision Dominant, and Vision Only versions of the same question. Text Dominant and Text Lite use the same image. In the text, the necessary conditions for solving the problem are highlighted in red, while redundant descriptive conditions are highlighted in blue. In the Vision Only version, the question is rendered in the image, with no textual format.

selected, as shown in Figure 21-26. As for the answer and analysis, each shape has a set of templates for different types of questions to choose from, as shown in Section A.4.1.

To further enhance the model's understanding of different forms of questions and better utilize the diverse modal information in the text and images, we divided the plain geometry questions generated by the Data Engine into four versions referring to MathVerse (Zhang et al., 2024b): Text Dominant, Text Lite, Vision Dominant, and Vision Only.

**Text Dominant**      We marked all the conditions required for solving the problem in the diagram and also described these conditions in the text, along with some redundant descriptive text.

**Text Lite**      All the conditions required for solving the problem are randomly divided into two parts: one part is marked in the diagram, and the other part is described in the text. In other words, the conditions in the diagram and the conditions in the text do not overlap.

**Vision Dominant**      All the conditions required for solving the problem are marked in the diagram, while the text only contains the question without any conditions.

**Vision Only**      Not only are all the conditions required for solving the problem marked in the diagram, but the question is also rendered in the diagram, leaving the text portion empty.

The differences among the four versions of the same question are illustrated in Figure 20. Each basic shape will retain a set of redundant conditions. During the shape generation process, there is a 50% probability of including these redundant conditions.

### ✏️ perimeter

What is the perimeter of {shape_id}?
How long is the perimeter of {shape_id}?
What is the total length of the perimeter of {shape_id}?
Can you calculate the perimeter of {shape_id}?
How do you find the perimeter of {shape_id}?
What is the measurement of the perimeter of {shape_id}?
What is the total perimeter measurement of {shape_id}?
Could you determine the perimeter of {shape_id}?
What would the perimeter of {shape_id} be?
What is the perimeter length of {shape_id}?
How would you calculate the perimeter of {shape_id}?
What's the perimeter of {shape_id}?
How much is the perimeter of {shape_id}?
Can you tell me the perimeter of {shape_id}?
What is the total boundary length of {shape_id}?
What is the entire perimeter length of {shape_id}?
How do you measure the perimeter of {shape_id}?
How would you determine the perimeter of {shape_id}?
What is the full perimeter of {shape_id}?
What is the overall perimeter of {shape_id}?
How can we calculate the perimeter of {shape_id}?

Figure 21: Perimeter problem templates.

### 📖 area

What is the area of the entire shape {shape_id}?
How much is the area of the entire shape {shape_id}?
What is the total area of the shape {shape_id}?
Can you calculate the area of the entire shape {shape_id}?
What is the overall area of the shape {shape_id}?
How do you find the area of the entire shape {shape_id}?
What is the measurement of the total area of {shape_id}?
What is the total area measurement of the shape {shape_id}?
Could you determine the area of the entire shape {shape_id}?
What would the area of the entire shape {shape_id} be?
What is the area size of the entire shape {shape_id}?
How would you calculate the total area of the shape {shape_id}?
What's the area of the entire shape {shape_id}?
How much area does the shape {shape_id} cover?
Can you tell me the area of the whole shape {shape_id}?
What is the overall area measurement of the shape {shape_id}?
What is the full area of the shape {shape_id}?
How do you calculate the area of the shape {shape_id}?
What is the total surface area of the shape {shape_id}?
How can we determine the area of the entire shape {shape_id}?
What is the total area of shape {shape_id}?

Figure 22: Area problem templates.

### △ base length

What is the length of the base {base_side_id} in isosceles triangle {shape_id}?
Can you tell me the length of the base {base_side_id} in the isosceles triangle {shape_id}?
What is the measurement of the base {base_side_id} in the isosceles triangle {shape_id}?
How long is the base {base_side_id} in the isosceles triangle {shape_id}?
What is the base length {base_side_id} in the isosceles triangle {shape_id}?
Could you provide the length of the base {base_side_id} in the isosceles triangle {shape_id}?
Can you specify the length of the base {base_side_id} in the isosceles triangle {shape_id}?
I need to know the length of the base {base_side_id} in the isosceles triangle {shape_id}.
Please tell me the length of the base {base_side_id} in the isosceles triangle {shape_id}.
What is the length of the side {base_side_id} that forms the base of the isosceles triangle {shape_id}?

Figure 23: Base length problem templates.

FUNCTION.    All functions will be examined with two types of questions: finding the derivative and finding the extrema. After obtaining the derivative, we calculate whether the derivative has zeros within the given domain. The presence of zeros determines the method for calculating the extrema.

# ✎ angle

In isosceles triangle {shape_id}, what is the measure of angle {bottom_angle_0} and angle {bottom_angle_1}?

In isosceles triangle {shape_id}, what is the measure of angle {bottom_angle_0} and angle {bottom_angle_1}?

What are the measures of angle {bottom_angle_0} and angle {bottom_angle_1} in the isosceles triangle {shape_id}?

Can you tell me the measures of angle {bottom_angle_0} and angle {bottom_angle_1} in the isosceles triangle {shape_id}?

In the isosceles triangle {shape_id}, what are the measures of angle {bottom_angle_0} and angle {bottom_angle_1}?

What is the measurement of angle {bottom_angle_0} and angle {bottom_angle_1} in the isosceles triangle {shape_id}?

How large are angle {bottom_angle_0} and angle {bottom_angle_1} in the isosceles triangle {shape_id}?

Please provide the measures of angle {bottom_angle_0} and angle {bottom_angle_1} in the isosceles triangle {shape_id}.

What are the measures of angles {bottom_angle_0} and {bottom_angle_1} in the isosceles triangle {shape_id}?

Can you specify the measures of angle {bottom_angle_0} and angle {bottom_angle_1} in the isosceles triangle {shape_id}?

In the isosceles triangle {shape_id}, what are the measurements of angle {bottom_angle_0} and angle {bottom_angle_1}?

Figure 24: Angle problem templates.

# ↻ arc length

In sector {shape_id}, what is the length of arc {arc_id}?
What is the length of arc {arc_id} in sector {shape_id}?
Can you tell me the length of arc {arc_id} in sector {shape_id}?
In the sector {shape_id}, what is the measurement of arc {arc_id}?
How long is arc {arc_id} in sector {shape_id}?
Please provide the length of arc {arc_id} in sector {shape_id}.
What is the arc length {arc_id} in the sector {shape_id}?
Could you specify the length of arc {arc_id} in sector {shape_id}?
What is the measurement of arc {arc_id} in the sector {shape_id}?
In sector {shape_id}, how long is arc {arc_id}?

Figure 25: Arc length problem templates.

# ▭ extend side length

In shape {shape_id}, what is the length of {extend_side_id}?
What is the length of {extend_side_id} in shape {shape_id}?
Can you tell me the length of {extend_side_id} in shape {shape_id}?
In the shape {shape_id}, what is the measurement of {extend_side_id}?
How long is {extend_side_id} in shape {shape_id}?
Please provide the length of {extend_side_id} in shape {shape_id}.
What is the side length {extend_side_id} in the shape {shape_id}?
Could you specify the length of {extend_side_id} in shape {shape_id}?
What is the measurement of {extend_side_id} in the shape {shape_id}?
In shape {shape_id}, how long is {extend_side_id}?

Figure 26: Extend side length problem templates.

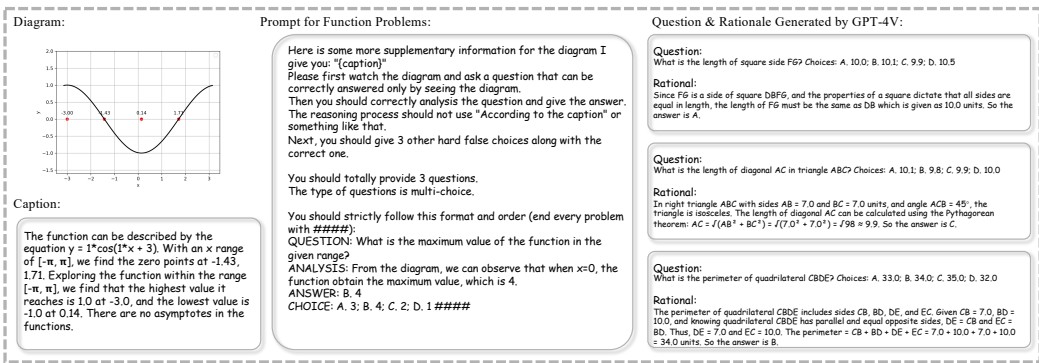

Figure 27: The function prompt for GPT-4V and the generated questions and rationals.

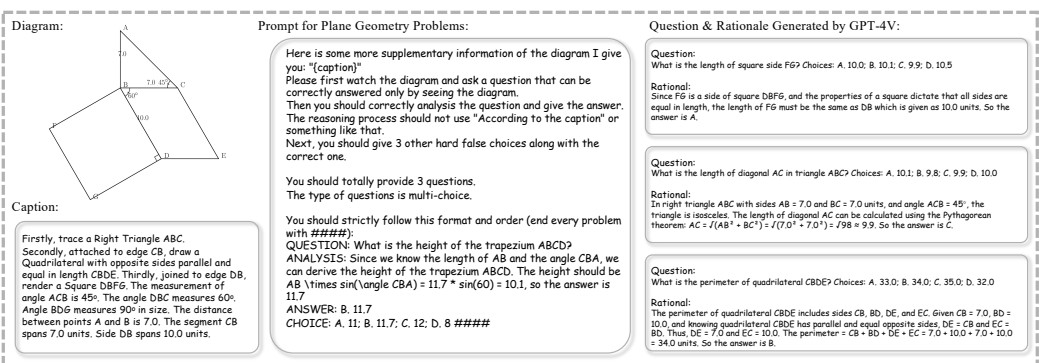

Figure 28: The geometry prompt for GPT-4V and the generated questions and rationals.

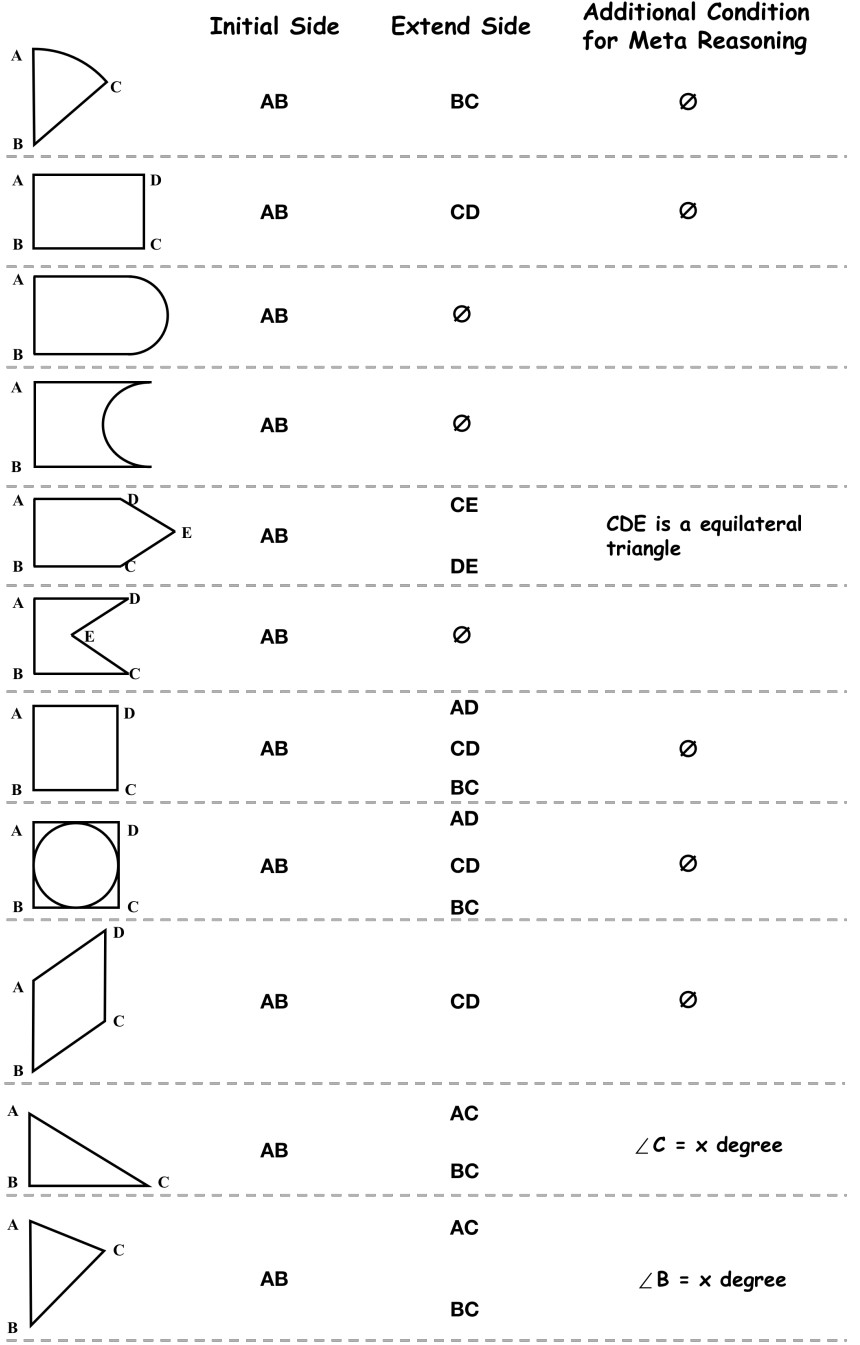

Figure 29: The initial side, extend side, and additional conditions for meta-reasoning of each basic shape. Some special shapes are not extended and only appear in the last position of the generation sequence, thus their extend side is $\emptyset$.

