**Q:** AB is tangent to circle O at point B, and the extended line of AO intersects circle O at point C. Connect BC, if angle A = 36.0, then angle C is equal to ?

**A:** Since AB is tangent to circle O at point B, we have OB perpendicular to AB. Therefore, angle ABO is 90°. Angle AOB is equal to 90° - angle A, which is 90°- 36°, giving us 54°. Since OC is equal to OB, angle C is equal to angle OBC. And since angle AOB is equal to angle C plus angle OBC, we have angle C equal to half of angle AOB, which is half of 54°, giving us 27°.

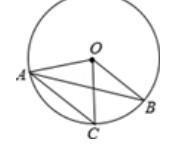

**Q:** in circle O, chord AC parallel radius OB, angle BOC = 50.0, the degree of angle OAB is ?

**A:** Since AC is parallel to OB, therefore angle BOC is equal to angle ACO which is 50 degrees. Also, OA is equal to OC, so angle OAC is equal to angle ACO which is 50 degrees. Since angle CAB is half of angle BOC, it is 25 degrees. Therefore, angle BAO is equal to angle OAC minus angle CAB which is 25 degrees.