# OpenReview forum: "MAVIS: Mathematical Visual Instruction Tuning with an Automatic Data Engine"
_ICLR.cc/2025/Conference — ICLR 2025 Poster_

### Official Review · Reviewer_3ZYS · 2024-11-01

**Soundness:** 3
**Presentation:** 3
**Contribution:** 3
**Rating:** 8
**Confidence:** 5

**Summary:**

This paper proposes MAVIS, a mathematical instruction tuning framework, including an automatic data generation engine, two multimodal mathematical datasets, and a specialized model MAVIS-7B. The mathematical instruction tuning of MAVIS consists of four progressive steps from visual pretraining to preference alignment, respectively targeting different issues in this field. The rule-based data engine can generate numerous mathematical data without costly human and GPT engagement, which contributes to sufficient data resources for training. The model MAVIS-7B shows good results on different benchmarks.

**Strengths:**

1. This paper comprehensively investigates different aspects of training multimodal math models, providing good insights in this field. The scope includes data curation, visual encoding, training pipelines, and learning methodologies. The exploration of better math-specific visual encoding, Math-CLIP, is interesting. This is a good reference for future work.

2. The proposed data engine is novel and of great value for obtaining large-scale training data. As discussed in the paper, the collection and annotation are expensive for multimodal math, while this data engine effectively alleviates this point. The author also shows detailed human study and comparison to ensure the data quality.

3. The MAVIS-7B model shows competitive performance in different benchmarks. The results validate the effectiveness and generalizability of the proposed data and training method.

**Weaknesses:**

1. The data engine is a rule-based system with fixed language templates. How to ensure the diversity of captions and questions in training data? It is very important to use wide and diverse data for robust model training.

2. The author only provides the results of the 7B model. What if the model size is larger, for example, 13B?

3. The author adopts introduced data and methods to train MAVIS from scratch. However, it is unknown whether these approaches are generalized enough to help a pre-trained large model, for example, LLaVA, to improve the mathematical performance.

**Questions:**

See weakness.

---

> ### Author Response · Authors · 2024-11-21
> **Response to Reviewer 3ZYS**
>
> We sincerely appreciate your insightful review and recognition of our work. We have provided detailed responses to your comment and updated the relevant content in the revised manuscript, hoping to address your concerns.
>
> > **Q1: As the data engine is a rule-based system, how to ensure the diversity of captions and questions in training data?**
>
> Thanks for pointing it out! we ensure the diversity of training data in two aspects:
> 1. **GPT-generated Diverse Templates.** While the construction of our captions and questions follows a template-based approach, we leverage GPT-4 to generate a broad range of diverse language templates. These templates are designed to include various linguistic expressions, syntactic structures, and phrasing styles to minimize repetitive patterns. In this way, we ensure that the resulting dataset covers a rich diversity of language styles, enhancing the robustness of the training data and enabling the model to generalize better across different tasks.
>
> 2. **Diversity through Real-world Data Integration.** We have incorporated 201K (24%) real-world collected and augmented data alongside the synthetic data (633K) into MAVIS-Instruct to diversify its distribution. Therein, 83K problems were manually collected and augmented from publicly available sources (as detailed in Appendix A.4.3), spanning diverse topics. The other 118K problems were sourced from existing mathematical datasets, predominantly covering plane geometry. Such real-world data well enriches the diversity of MAVIS-Instruct data, generalizing MLLMs across various mathematical tasks.
>
> > **Q2: The author only provides the results of the 7B model. What if the model size is larger, for example, 13B?**
>
> Thanks for your advice! In the table below, we present the results of the MAVIS model using two different sizes of LLMs: Mammoth-2-7B and Mammoth-2-8x7B. The accuracy on MathVerse demonstrates that with larger LLMs, the MAVIS model showcases good scaling performance and achieves enhanced mathematical reasoning capabilities.
> Model|LLM Size|All|Text Dominant|Text Lite|Vision Intensive|Vision Dominant|Vision Only
> :-|:-|:-:|:-:|:-:|:-:|:-:|:-:|
> MAVIS |7B |28.4 | 41.6 |29.5 | 27.9 | 24.7 | 18.3
> MAVIS |8x7B |**29.4** |**42.3** | **30.8** | **28.9** | **25.6** | **19.5**
>
>
> > **Q3: Whether these approaches can help a pre-trained large model (LLaVA) improve the mathematical performance?**
>
> Thanks for your suggestion! Our curated data and training techniques can well generalize to other pre-trained MLLM to enhance their mathematical performance. In the table below, on top of LLaVA-NeXT-8B, we progressively employ our MAVIS-Instruct for instruction tuning and DPO alignment, both training for one epoch with an learning rate as 1e$^{-5}$. The two continual training stages significantly boost the mathematical problem-solving skills of LLaVA-NeXT.
>
> Model|LLM Size|All|Text Dominant|Text Lite|Vision Intensive|Vision Dominant|Vision Only
> :-|:-|:-:|:-:|:-:|:-:|:-:|:-:|
> LLaVA-NeXT|8B |15.6 | 19.4 | 15.2 | 16.8 | 15.2 |11.3
> w MAVIS-Instruct|8B |22.8 | 32.3 | 25.3 | 24.6 | 18.3 |14.2
> w DPO|8B |**24.0** | **33.7** | **26.9** | **25.4** | **19.1** |**15.1**

---

> > ### Comment · Reviewer_3ZYS · 2024-11-26
> >
> > Thank you for your thoughtful rebuttal. I maintain my rating.

---

> > > ### Author Response · Authors · 2024-11-27
> > > **Thanks for your recognition of our rebuttal!**
> > >
> > > Dear Reviewer 3ZYS,
> > >
> > > Thanks for your thoughtful acknowledgment of our rebuttal and efforts. We greatly appreciate your insightful comments, which have been instrumental in enhancing the quality of our work.
> > >
> > > Regards,
> > >
> > > Authors

---

### Official Review · Reviewer_zbps · 2024-11-02

**Soundness:** 3
**Presentation:** 4
**Contribution:** 3
**Rating:** 6
**Confidence:** 4

**Summary:**

This paper introduce MAVIS, a multimodal large language model trained with mathmatical dataset generated by data engine.
Specifically, the authors constructed MAVIS-caption to train CLIP-Math with better mathmatical visual representation ability and MLP multimodal connector. Then the authors constructed MAVIS-Instruct to endow MLLM instruct follow and CoT reasoning ability.
The evaluation result shows that MAVIS achieve good performance on multiple math-domain benchmarks.

**Strengths:**

- The writing is clear and easy to follow.
- The idea of using dataset generation engine to enhance MLLM math visual reaoning ability is straightforward and effective.
- The training stretagy, including the staged training, and the dataset design are well-established.

**Weaknesses:**

- Although MAVIS is a mathematical visual specialist, the performance is still similar with some generalist MLLMs. Especially on vision-only version of Mathverse, where MAVIS is only 0.9% better than LLaVA-NEXT-110B. Suggesting that the MAVIS's visual representation learning is not very effective.
- The generation engine can only cover a limited set of mathmatical diagrams. For example, circumcircle of a triangle, which is a very common geometry shape, can not be generated by the engine. Since the engine is manually designed, it will be challenging to cover all kinds of math visual diagram types. If the goal is to train a generalize math MLLM, discussion on the generalize ability of MAVIS is needed, since the training shape is limited.

**Questions:**

- I'm wondering the result of Table 6, Mathverse but on vision only version. Since CLIP-math helps the model improve its visual representation, a performance improvement on vision-only version can be more convincing.

---

> ### Author Response · Authors · 2024-11-21
> **Response to Reviewer zbps (Part 1)**
>
> We sincerely appreciate your valuable comments and recognition of our work. We have provided detailed responses to your comment and updated the relevant content in the revised manuscript, hoping to address your concerns.
>
> > **Q1: Although MAVIS is a mathematical visual specialist, the performance is still similar with some generalist MLLMs. For example, on the vision-only version of MathVerse, MAVIS is only 0.9% better than LLaVA-NeXT-110B**
>
> Thanks for pointing this out! We would like to clarify this observation and address it from several perspectives:
>
> 1. **7B vs. 110B.** Our model is built on an LLM with ***only 7B parameters***, whereas LLaVA-NeXT-110B utilizes a significantly larger LLM with ***110 billion parameters***, over 10 times the size of ours. Despite this disparity, we believe it is a ***noteworthy achievement*** that, our 7B model surpasses a 110B model in the overall results of MathVerse with significant margins (***+6.8 CoT scores and +3.9 accuracy***). Compared to other models of comparable sizes, e.g., LLaVA-NeXT-8B, MAVIS achieves consistently superior superior results with +18 CoT scores and +12.8 accuracy. This highlights the effectiveness of our training strategies and curated datasets in maximizing the performance of a smaller model.
>
> 2. **Better CoT Performance on Vision-only Problems.** While MAVIS-7B’s accuracy score on vision-only problems is slightly lower than LLaVA-NeXT-110B (-2.3), our model achieves a higher CoT evaluation score with a +0.9 margin. ***The CoT score is a strong indicator of step-by-step reasoning capabilities***. Thus, this advantage demonstrates the effectiveness of our rationale annotation and DPO training to conduct high-quality CoT reasoning.
>
>
> 3. **Improving Vision-only Results with OCR Data.** Vision-only problems in MathVerse require the model to interpret question texts rendered directly in diagrams, relying heavily on OCR capabilities. MAVIS-7B, however, was not trained with OCR-specific datasets, limiting its performance in these tasks. In contrast, generalist models like LLaVA-NeXT include extensive OCR datasets such as OCRVQA, DocVQA, and SynDog-EN, which boosts their OCR capabilities and performance in vision-only problems. To address this gap, we also include OCR datasets (OCRVQA and DocVQA) in our third-stage instruction tuning to enhance OCR capabilities. The results, as shown in the table below, reveal ***a notable improvement in vision-dominant and vision-only problems***. This highlights the potential of MAVIS-7B to achieve even higher results with better OCR integration.
>
>
> Model|LLM Size|All|Text Dominant|Text Lite|Vision Intensive|Vision Dominant|Vision Only
> :-|:-|:-:|:-:|:-:|:-:|:-:|:-:|
> LLaVA-NeXT|8B |15.6 | 19.4 | 15.2 | 16.8 | 15.2 |11.3
> LLaVA-NeXT|110B| 24.5 | 31.7 | 24.1 |21.0| 22.1 | 20.7
> MAVIS-7B |7B |28.4 | **41.6** |**29.5** | **27.9** | 24.7 | 18.3
> MAVIS-7B w OCR |7B |**28.9** | 40.8 | 29.2 | 27.4 | **26.2** | **21.1**

---

> > ### Author Response · Authors · 2024-11-21
> > **Response to Reviewer zbps (Part 2)**
> >
> > > **Q2: Since the engine is manually designed, it will be challenging to cover all kinds of math diagrams. The discussion on the generalization ability of MAVIS is needed**
> >
> > Thanks for pointing this out! We would like to illustrate the generalization capability of MAVIS in three aspects:
> > 1. **Diversity through Real-world Data Integration.** We have incorporated 201K (24%) real-world collected and augmented data alongside the synthetic data into MAVIS-Instruct to diversify its distribution. Therein, 83K problems were manually collected and augmented from publicly available sources (as detailed in Appendix A.4.3), spanning diverse topics. The other 118K problems were sourced from existing mathematical datasets, predominantly covering plane geometry. Such real-world data well enriches the diversity of MAVIS-Instruct data, generalizing MLLMs across various mathematical tasks.
> >
> > 2. **Validation Across Diverse Benchmarks.**
> > As shown in Table 4 of the paper, MAVIS demonstrates robust performance across a range of benchmarks, encompassing varied mathematical scenarios: 1) ***Subjects***: plane geometry (GeoQA), solid geometry (MathVerse), functions (FunctionQA), and algebra (MathVision); 2) ***Difficulty levels***: elementary-school (MathVista), high-school (MathVerse), competition-level (MathVision), and college-level (MMMU-Math); 3) ***Diagrams***: geometry (GeoQA), functions (FunctionQA), and statistics (MathVision); and 4) ***Capabilities***: visual perception (MathVerse), CoT reasoning (MathVerse), and knowledge comprehension (We-Math).
> > Additionally, we provide examples for circumcircles of triangles in ***Figure 11 of the Appendix*** in the revised manuscript, showcasing MAVIS-7B's ability to solve such mathematical problems.
> >
> > 3. **Impact of Real-World and Synthetic Data.** In the table below, we conduct an ablation study to investigate the impact of two data subsets, excluding the DPO training for fairness. As shown, the two sources of data exhibit ***complementary characteristics***, and ***the integration of them contributes to the best generalizability***. Synthetic data can better improve the results of FunctionQA and MMMU-Math, since they both contain a large portion of function problems. In contrast, real-world data contributes more to GeoQA, which are more aligned in geometry problems.
> > Synthetic Data | Real-world Data | MathVerse Acc| GeoQA | FunctionQA | MMMU-Math
> > :-:|:-:|:-:|:-:|:-:|:-:
> > ✔️ | -- | 22.6|44.2|37.1|34.6
> > -- | ✔️ |24.3|66.4|25.8|29.8
> > ✔️ | ✔️ |27.5|66.7|40.3|39.2
> >
> > > **Q3: A performance improvement on vision-only version can be more convincing**
> >
> > Thanks for the suggestion! Please refer to ***Q1 of this rebuttal***, where we discuss incorporating OCR datasets (e.g., OCRVQA and DocVQA) during instruction tuning. This significantly enhances MAVIS-7B’s performance on vision-only problems.

---

> ### Comment · Reviewer_zbps · 2024-11-25
>
> Thanks authors for their detailed response, I'm generally satisfied and will keep my positive rating.
>
> I appreciate the author's effort in improving the vision-only performance by incorporating OCR training, the performance, however, only improves about 3%. This might suggest that OCR ability is benificial but not the main bottleneck of their relatively weak vision-only performance. I know it is challenging to address this during this period, but it would be helpful to provide some qualitative examples to give us some intuition about what are the main bottlenecks.

---

> > ### Author Response · Authors · 2024-11-27
> > **Thanks for your recognition of our rebuttal!**
> >
> > Dear Reviewer zbps,
> >
> > Thank you for acknowledging our rebuttal and efforts! We deeply appreciate your insightful comments, which have been invaluable in helping us improve our work.
> >
> > As you suggested, in ***Figure 12 of the revised paper***, we present qualitative examples of vision-only problems tackled by MAVIS-7B with OCR training. Our observations indicate that while incorporating OCR data has led to a +3% performance improvement, the primary bottleneck remains the limited perception accuracy for rendered questions and numerical values within diagrams. This highlights a significant improvement space for OCR capabilities in mathematical elements.
> >
> > We recognize this as an important direction for future work in exploring more effective strategies for integrating OCR data during mathematical training to further enhance vision-only performance.
> >
> > Thanks again for your constructive feedback.
> >
> > Regards,
> >
> > Authors

---

### Official Review · Reviewer_uZLE · 2024-11-03

**Soundness:** 3
**Presentation:** 3
**Contribution:** 2
**Rating:** 6
**Confidence:** 4

**Summary:**

This work primarily aims to enhance the visual mathematical reasoning capabilities of MLLMs. The authors observe that existing visual encoders struggle to capture abstract information in math diagram, and MLLMs lack strong visual mathematical reasoning abilities. To address this, they propose a data collection pipeline and collected approximately 1.4 million math diagram-caption pairs and VQA with CoT data. By fine-tuning the CLIP visual encoder, aligning CLIP with the LLM, and applying Supervised Fine-Tuning (SFT) and Direct Preference Optimization (DPO) alignment techniques, they train a 7B math MLLM.

**Strengths:**

- The motivation behind this work is clear. For example, it identifies the semantic gap between natural images with rich colors and textures and mathematical diagrams with grayscale and abstract elements. Therefore, constructing math diagram-caption pairs is necessary to enhance visual embeddings.
- The data collection pipeline does not require extensive manual annotation or the use of OpenAI API, it is rule-based and automated, making it relatively low-cost.
- This work involves a substantial amount of engineering effort, collecting 1.4 M data in the mathematical domain. The model also achieved high performance across multiple benchmarks.

**Weaknesses:**

- The main contribution of this paper is the proposal of an automated data collection pipeline that doesn’t require extensive manual annotation or the use of the OpenAI API. However, the paper lacks an overview figure in the main text, which may make it challenging for readers to grasp the key idea of this data collection pipeline.
- This work lacks a certain degree of novelty. For instance, it does not propose a new model architecture, and the training approach is a combination of the popular methods, such as CLIP’s contrastive training[1], LLaVA’s visual instruction tuning, and Direct Preference Optimization[3] (DPO). The model and training method are not specifically optimized for the mathematical domain; instead, they rely on a general model and training approach, with only a mathematical dataset provided for fine-tuning. (Fortunately, the engineering effort is substantial.)

[1] Radford A, Kim J W, Hallacy C, et al. Learning transferable visual models from natural language supervision[C]//International conference on machine learning. PMLR, 2021: 8748-8763.

[2] Liu H, Li C, Wu Q, et al. Visual instruction tuning[J]. Advances in neural information processing systems, 2024, 36.

[3] Rafailov R, Sharma A, Mitchell E, et al. Direct preference optimization: Your language model is secretly a reward model[J]. Advances in Neural Information Processing Systems, 2024, 36.

**Questions:**

- In Section 3.2, why only freeze the CLIP visual encoder and train the projection layer along with the LoRA-based LLM? In LLaVA [1], both the visual encoder and LLM are frozen, with only the projection layer trained to align the visual encoder and LLM. If training the LoRA-based LLM is better, please provide a rationale and ablation experiments to support this approach.

- The core idea behind the rule-based caption and instruction data collection is akin to template filling, and the concept for constructing CoT data seems similar to deductive reasoning?

[1] Liu H, Li C, Wu Q, et al. Visual instruction tuning[J]. Advances in neural information processing systems, 2024, 36.

---

> ### Author Response · Authors · 2024-11-21
> **Response to Reviewer uZLE (Part 1)**
>
> We sincerely appreciate your insightful reviews and constructive advice. We have provided detailed responses to your comment and updated the relevant content in the revised manuscript, hoping to address your concerns.
>
> > **Q1: The paper lacks an overview figure, which may make it challenging for readers to understand the data collection pipeline.**
>
> Thanks for your suggestion! A clear overview figure can aid readers in better understanding the methodology and contributions of our proposed data engine. Accordingly, we have included an overview figure in ***Figure 2 of the revised manuscript***.
>
>
> > **Q2: This work does not propose a new model architecture or training approach, with only a mathematical dataset provided for fine-tuning. (Fortunately, the engineering effort is substantial)**
>
> Thanks very much for recognizing our engineering effort. We would like to clarify the main purpose and contributions of this study, which are summarized as follows:
>
> 1. **The Main Goal of this work is to provide a general and robust paradigm** for training MLLMs in mathematical reasoning, including training pipelines and datasets. Our focus is not on proposing new fancy model architectures or training approaches, but on exploring a ***concise and effective*** method to target specific challenges in this field. This work provides valuable insights and a practical foundation for future research in mathematical MLLMs.
> 2. **Our Training Pipeline is specifically designed for key shortcomings in math.** Given our observation of MLLMs' three issues (*visual encoding of math diagrams, diagram-language alignment, and chain-of-thought (CoT) reasoning*), we respectively devise four training stages to alleviate them. ***The first*** pre-training stage of Math-CLIP bridges the semantic gap from general images to math diagrams. ***The second*** cross-modal alignment stage enhances the interpretation of LLMs on diagram embeddings from Math-CLIP. ***The third*** mathematical instruction-tuning stage endows MLLMs with fundamental problem-solving and reasoning skills. ***The fourth*** DPO training further elicit advanced reasoning capabilities and generate high-quality CoT rationales.
> Therefore, while the techniques themselves are not new, our work represents ***the first effort*** to integrate and order them systematically for mathematical domains, collectively optimizing MLLM performance for mathematical reasoning, which provides reference for future work in this area.
> 3. **The Curation of Datasets is well tailored for the mathematical domain.** For each training stage we designed, we respectively support a corresponding dataset to fully unleash the training effectiveness, e.g., MAVIS-Caption for the first and second stage, MAVIS-Instruct for the third stage, and DPO ranking data for the fourth stage. Unlike conventional instruction-tuning datasets, ***our curation process incorporates the unique characteristics of mathematical tasks***. For example,
> to mitigate the high costs of manual data collection and annotation, we develop ***an automatic data engine*** to efficiently generate math problems, including diagrams, captions, questions, and CoT rationales.
> We refined these problems into ***text-lite versions*** by removing textual redundancy, ensuring the model focuses more effectively on diagram interpretation during training.
> These designs leverage domain-specific insights to enhance the training process for mathematical reasoning tasks.
>
> In conclusion, our work is far from a naive application of existing techniques. Instead, it integrates thoughtful designs in training methodologies and dataset curation, specialized for the multi-modal mathematical domain.

---

> > ### Author Response · Authors · 2024-11-21
> > **Response to Reviewer uZLE (Part 2)**
> >
> > > **Q3: In the alignment stage, why train the projection layer along with the LoRA-based LLM? In LLaVA, only the projection layer is trained. Ablation study is needed**
> >
> > Thanks for your advice! Unlike LLaVA, we train both the projection layer and the LoRA-based LLM during the alignment stage. This design is due to ***the difference between general visual tasks and mathematical tasks:***
> >
> > 1. **For General Visual Tasks (LLaVA)**, the LLM is ***normally required to generate daily natural language responses***, e.g., descriptive captions or instruction-following outputs. These outputs often rely on ***pre-existing knowledge within the pre-trained LLM***. Consequently, in LLaVA, there is no need to unfreeze the LLM to learn new types of outputs.
> >
> > 2. **For Mathematical Tasks (MAVIS)**, LLMs are required to ***generate math-specific responses***, e.g., geometric or functional descriptions, formula, and theorems. These outputs often involve ***new domain knowledge not inherent in pre-trained LLMs***. Given this, we add learnable LoRA layers to infuse new knowledge into LLMs, enhancing its ability to produce high-quality mathematical expressions. At the same time, we aim to prevent the LLM from overly fitting to diagram captioning tasks during alignment. Therefore. ***by using LoRA-based tuning, we preserve the LLM’s generalizable pre-trained language knowledge, while injecting specialized math-specific capabilities***.
> >
> > 3. As you suggested, we conduct an ablation study to compare different LLM training settings during alignment. We evaluate two tasks: the CIDEr score for diagram captioning on 100 validation samples (the same setting as Table 6 in the Appendix) and the accuracy score on MathVerse. The results indicate that, the LoRA-based manner performs the best, enabling MLLMs to generate high-quality math captions and preserving pre-trained knowledge for better problem-solving capabilities.
> > LLMs|Caption CIDEr | MathVerse Acc
> > :-|:-:|:-:
> > Frozen | 79.6|26.2
> > Unfrozen | 146.2|28.1
> > LoRA-based | 161.3 |28.4
> >
> > > **Q4: The caption and instruction data collection is akin to template filling, and the CoT construction seems similar to deductive reasoning?**
> >
> > Yes, just as you said.
> >
> > For captions and questions, we utilize diverse language templates generated by GPT-4, covering a wide range of linguistic variations, and concatenate them using our rule-based data engine.
> >
> > The CoT rationales are indeed built upon a deductive reasoning framework. Starting from an initial mathematical condition, we iteratively deduce intermediate steps with diverse templates to arrive at the final answer, making the reasoning process both natural and robust.

---

> > > ### Comment · Reviewer_uZLE · 2024-11-21
> > > **Raising the Rating: 5 → 6**
> > >
> > > Thank you for the thoughtful responses and revisions. The addition of the overview figure greatly improves the paper’s clarity. The detailed explanations and supplementary experiments effectively address my concerns, and I am raising my rating to 6.

---

> > > > ### Author Response · Authors · 2024-11-21
> > > > **Thanks for your recognition of our rebuttal!**
> > > >
> > > > Dear Reviewer uZLE,
> > > >
> > > > Thank you for acknowledging our rebuttal and efforts! We deeply appreciate your insightful comments, which have been invaluable in helping us improve our work.
> > > >
> > > >
> > > > Regards,
> > > >
> > > > Authors

---

### Official Review · Reviewer_dHAm · 2024-11-04

**Soundness:** 3
**Presentation:** 3
**Contribution:** 3
**Rating:** 6
**Confidence:** 4

**Summary:**

The paper argues that the existing models lack in mathematical reasoning in the visual contexts due to the lack of math diagram understanding by the vision encoders, limited diagram-language alignment data, and inaccurate capability of large models (e.g.,GPT-4V, Gemini-Pro) to perform chain-of-thought reasoning. To address this, the paper proposes MAVIS, a rule-based and automatic data engine used for visual instruction tuning of the 7B large multimodal models. Specifically, the paper focuses on plane geometry diagrams, analytic geometric diagrams, and function diagrams. Subsequently, the paper performs 4-stage training on MAVIS-caption and instructs to achieve good performance on the MathVerse dataset. While the data collection, and training pipelines look reasonable, the paper severely lacks solid experiments to establish the usefulness of the generated data.

**Strengths:**

- The dataset creation pipeline is quite rigorous ranging for the three categories – including caption and instruct data generation.
- The size of the captioning and instruct dataset is quite large, which is a plus too. The method for creating diagram-caption data using GPT-4 as template generators instead of caption generators is interesting.
- The paper performs 4-stage training: CLIP-Math, Aligning diagram-language alignment, instruction tuning and preference alignment.
- The final numbers showcase the benefits of their data and training on the evaluations.

**Weaknesses:**

- Mammoth-2 is supposedly a strong model for math reasoning for text. It remains unclear how much benefit does the MAVIS data give over finetuning Mammoth-2 with existing visual instruction tuning dataset (e.g., LLaVA data). This is crucial to control for the choice of the base model itself.
- The main result is on the MathVerse dataset which consists of author-created solid geometry and function problems and plane geometry problems sourced from GeoQA and Geometry3K. The authors augment their dataset with the problems from the same dataset in L338. It is crucial to see if most of the improvements in Table 2 are in the plane geometry, while solid geometry and functions still suffer.
- Related to above: there is a synthetic component to the dataset and a real-data (e.g., Geometry datasets and sourcing problems from internet) component to the instruct data. How much of the performance comes from the synthetic data and how much of it is contributed by adding the real data is unclear.
- There are no experiments to study data scaling — does training with all the data better than training with 25%, 50% and 75% of the entire dataset. If the data quality is high, then the model performance should scale with data.
- Table 4 has too many dashes for open-source MLLMs which makes it hard to put the model improvements in the complete context. The table makes it look as if some numbers have been skipped selectively.
- The public sources (or protocol) for selecting the math problems from public sources in L322 is unclear. There is no data contamination study to understand the overlap between the generated data and evaluation data.
- Minor: Table 3 and 4 adds a * on the models that are consider math specialist but there is no * on the MAVIS models themselves. I would consider them math specialists too.

**Questions:**

mentioned in the weakness

---

> ### Author Response · Authors · 2024-11-21
> **Response to Reviewer dHAm (Part 1)**
>
> We sincerely appreciate your insightful reviews and constructive advice. We have provided detailed responses to your comment and updated the relevant content in the revised manuscript, hoping to address your concerns.
>
> > **Q1: It remains unclear how much benefit of MAVIS data over existing visual instruction data (e.g., LLaVA data) in fine-tuning Mammoth-2**
>
> Thanks for your advice! To investigate the detailed benefit of MAVIS data in Mammoth-2 (7B), we conduct an ablation study as below and updated it in the revised manuscript. As you suggested, we adopt the data from LLaVA-NeXT (558K for pre-training and 760K for fine-tuning) and compare with our MAVIS data (558K MAVIS-Caption for pre-training and 834K MAVIS-Instruct for fine-tuning). We evaluate the accuracy metric on MathVerse and don't involve the DPO training stage. Result in the first row denotes the original LLaVA-NexT with LLaMA-3 (8B).
>
> Visual encoder | LLM | Pre-training | Fine-tuning | MathVerse Acc
> -|-|-|-|:-:
> CLIP | LLaMA-3 | LLaVA data | LLaVA data | 15.6
> CLIP | Mammoth-2 | LLaVA data | LLaVA data | 18.3
> CLIP | Mammoth-2 | LLaVA data | **MAVIS-Instruct** | 25.7
> CLIP | Mammoth-2 | **MAVIS-Caption** | **MAVIS-Instruct** | 26.4
> **Math-CLIP** | Mammoth-2 | **MAVIS-Caption** | **MAVIS-Instruct** | 27.5
>
> From the results, we observe:
> 1. Mammoth-2 brings +2.7 score compared to LLaMA-3 with, indicating its prior knowledge of mathematical problem solving.
> 2. Applying MAVIS-Instruct for fine-tuning can significantly enhance the performance by +7.4, demonstrating the great advantage of our data for mathematical reasoning compared to general visual instruction data.
> 3. Using MAVIS-Caption for pre-training and training Math-CLIP encoder can further attain higher scores with enhanced mathematical visual perception. ***Therefore, our MAVIS data contributes to +9.2 score in total over Mammoth-2 with LLaVA data.***
>
> ---
> > **Q2: It is crucial to see if most of the improvements in MathVerse are mainly in the plane geometry, while solid geometry and functions still suffer**
>
> 1. Sorry for the confusion caused. Our MAVIS data (including MAVIS-Caption and -Instruct) mainly consists of two subjects: **plane geometry (985K, 69%)** and **functions (437K, 31%)**. They are expected to improve MLLM with their solving capabilities of both geometry and function problems.
>
> 2. We provide the detailed subject scores of MAVIS-7B on MathVerse, and compare the CoT evaluation score (the subject-level accuracy score is not released) with other models from the official leaderboard.
>
> Model | All (CoT-Eval) | Plane Geometry | Solid Geometry | Functions
> -|:-:|:-:|:-:|:-:
> LLaVA-NeXT | 17.2 | 15.9 | 19.6 | 23.1
> ShareGPT4V | 17.4 |16.9 | 15.0 | 20.2
> SPHINX-MoE | 22.8 | 24.5 | 15.8 | 19.5
> InternLM-XC2. | 25.9 | 26.2 | 20.1 | 23.7
> **MAVIS-7B** | 35.2 | 37.1 | 28.9 | 31.0
>
> The results showcase that ***our model achieves leading performance on all three subjects***. Therein, the proficiency of plane geometry and functions are primarily due to the training with our curated MAVIS data. For solid geometry, as it is similar to plane geometry in both visual appearance and reasoning process, we believe our model can well generalize its learned knowledge and reasoning capabilities to the solid geometry domain for improved performance.
>
> ---
> > **Q3: How much of the performance comes from the synthetic data and the real data is unclear.**
>
> Thanks for your suggestion! In MAVIS-Instruct, we integrate both synthetic problems from the data engine (633K, 76%) and real-world problems augmented with GPT (201K, 24%). Synthetic data is composed of both geometry and functions, while real-world data mainly focuses on geometry. We provide the ablation of data component below and don't involve the DPO training stage for fairness.
>
> Synthetic Data | Real-world Data | MathVerse Acc| GeoQA | FunctionQA | MMMU-Math
> :-:|:-:|:-:|:-:|:-:|:-:
> ✔️ | -- | 22.6|44.2|37.1|34.6
> -- | ✔️ |24.3|66.4|25.8|29.8
> ✔️ | ✔️ |27.5|66.7|40.3|39.2
>
> As shown, the two sources of data exhibit ***complementary characteristics***, and ***both of them are significant to the final performance***. Synthetic data can better improve the results of FunctionQA and MMMU-Math, since they both contain a large portion of function problems. In contrast, real-world data contributes more to GeoQA, which are more aligned in geometry problems.

---

> > ### Author Response · Authors · 2024-11-21
> > **Response to Reviewer dHAm (Part 2)**
> >
> > > **Q4: There are no experiments to study data scaling (25%, 50% and 75% of the entire dataset).**
> >
> > Thanks for pointing out! In the 834K MAVIS-Instruct, we randomly sample 25%, 50% and 75% data for instruction tuning without the DPO stage, and report the accuracy metric on MathVerse. From the table below, we observe the result of MAVIS-7B can well increase as the data is scaled up, indicating the promising potential of our data to further enhance mathematical reasoning capabilities.
> >
> > 25% | 50% | 75% | 100%
> > :-:|:-:|:-:|:-:
> > 23.3 | 25.7 | 26.9 | 27.5
> >
> > ---
> > > **Q5: Too many dashes in Table 4 makes it hard to put the model improvements in the complete context. It look as if some numbers have been skipped selectively.**
> >
> > Note that ***We Didn't Skip Any Results Selectively***. We have tried our best to collect results of different models from different benchmarks. The dashes are used only for the ***Publicly Unavailable Results*** from their official leaderboards.
> >
> > To better evaluate our model in the complete context, we manually test the missing results of several advanced models with 7B and 13B sizest using their official checkpoints. * denotes mathematical visual specialists. As showcased below, compared to previous MLLMs with different LLM sizes, our 7B model attains superior performance across different benchmarks.
> >
> > Model |LLM Size|GeoQA |FunctionQA |MMMU-Math |MathVision |GPS |ALG |GEO |S1 |S2| S3
> > :-|:-:|:-:|:-:|:-:|:-:|:-:|:-:|:-:|:-:|:-:|:-:
> > G-LLaVA*|13B|67.0| 24.2| 27.6 |1.3 |36.1 |24.6| 33.1| 32.4 |30.1 |32.7
> > Math-LLaVA*|13B|62.3 |38.7 |36.1 |15.5 |57.7 |53.0 |56.5 |37.5 |30.5 |32.4
> > InternLM-XC2. |7B|66.4|38.7| 30.1| 14.5 |63.0| 56.6 |62.3 |47.0| 33.1 |33.0
> > MAVIS-7B*|7B|68.3 |50.0| 42.4 |19.2 |64.1| 59.2| 63.2 |57.2| 37.9 |34.6
> >
> > ---
> > > **Q6: The sources and protocols for selecting public math problems are unclear, and there is no data contamination study for the overlap between generated and evaluation data.**
> >
> > In A.4.3 of the Appendix, we have provided some details of public math problem collection, e.g., data sources and processing methods. We detail the collection process below:
> > 1. From the three public sources, we collect each problem as complete as possible, including questions, diagrams, answers, category information, and rationales if available. The problems are in high-school levels, and mainly span plane geometry and functions (including analytic geometry).
> > 2. According to their categories (subject, subfield, and difficulty level), we divided them into different parts and demanded six expert annotators for a meticulous check of the correctness and integrity of problems. They were required to refine detailed CoT rationales, and ***ensure the problem is not overlapped with any evaluation data*** by visually comparing the diagrams. After this process, we obtain 4K problems in total.
> > 3. Finally, with 4K problems, we feed each problem into GPT-4V with customized prompt (Figure 15) to eliminate the redundant information within question text, constructing text-lite problems for better training of mathematical visual capabilities.
> >
> > ---
> > > **Q7: Table 3 and 4 add a * on math specialist models, but there is no * on the MAVIS model.**
> >
> > Thanks for pointing it out! We have added * on the MAVIS model in the revised manuscript.

---

> > > ### Comment · Reviewer_dHAm · 2024-11-23
> > > **Response to the authors**
> > >
> > > Hi,
> > >
> > > I thank the authors for running these experiments, the new results look good and improve the quality of the paper.

---

> > > > ### Author Response · Authors · 2024-11-23
> > > > **Thanks for your recognition of our rebuttal!**
> > > >
> > > > Dear Reviewer dHAm,
> > > >
> > > > Thank you for acknowledging our rebuttal and efforts! We deeply appreciate your insightful comments, which have been invaluable in helping us improve our work.
> > > >
> > > > Regards,
> > > >
> > > > Authors

---

### Meta-Review · Area_Chair_CnFp · 2024-12-15

**Metareview:**

This paper introduces MAVIS, a multimodal instruction tuning framework designed to enhance mathematical reasoning capabilities in large multimodal language models (MLLMs). It features an automatic data engine that generates mathematical datasets, including MAVIS-Caption and MAVIS-Instruct, without requiring manual annotations. The proposed training pipeline addresses visual encoding, alignment, and chain-of-thought (CoT) reasoning. Experiments demonstrate MAVIS-7B's significant improvements across multiple benchmarks, including MathVerse and GeoQA. While reviewers appreciated the dataset's scale, training strategy, and competitive results, concerns were raised about limited generalization, constrained mathematical diagram diversity, and the computational limitations of the rule-based engine. These issues were addressed through additional experiments and clarifications during the rebuttal phase. Overall, the work is recommended for Accept (poster) due to its substantial contributions to mathematical visual reasoning in MLLMs.

**Additional Comments On Reviewer Discussion:**

The reviewers raised concerns about generalization capabilities, the rule-based nature of the data engine, and comparisons with larger models. The authors responded comprehensively, adding ablation studies, extending evaluations with OCR datasets, and showcasing MAVIS's scalability to larger LLMs. Notably, improvements on vision-only tasks and the integration of real-world data strengthened the paper’s contributions. Although some concerns about the engine's extensibility remain, the rebuttal effectively addressed key issues, leading to positive adjustments in scores and a consensus toward acceptance.

---

### Decision · Program_Chairs · 2025-01-22

Accept (Poster)